# Interpretable time series analysis with Gumbel dynamics

## Abstract

Switching dynamical systems can model complicated time series data while maintaining interpretability by inferring a finite set of dynamics primitives and explaining different portions of the observed time series with one of these primitives. However, due to the discrete nature of this set, such models struggle to capture smooth, variable-speed transitions, as well as stochastic mixtures of overlapping states (e.g., non-instantaneous state transitions), and the inferred dynamics often display spurious rapid switching on real-world datasets. Here, we propose the Gumbel Dynamical Model (GDM). First, by introducing a continuous relaxation of discrete states and a different noise model defined on the relaxed-discrete state space via the Gumbel distribution, GDM expands the set of available state dynamics, allowing the model to approximate smoother and non-stationary ground-truth dynamics more faithfully. Breaking from established literature, this new class directly links states to observations and does not blur latent dynamics with Gaussian noise. Second, the relaxation makes the model fully differentiable, enabling fast and scalable training with standard gradient descent methods. We validate our approach on standard simulation datasets and highlight its ability to model soft, sticky states and transitions in a stochastic setting. Furthermore, we apply our model to two real-world datasets, demonstrating its ability to infer interpretable states in stochastic time series with multiple dynamics, a setting where traditional methods often fail.

## 1 Introduction

Natural behaviors give rise to complex time series data with non-stationary and nonlinear dynamics. Such dynamical phenomena are often well approximated within a temporal neighborhood by a small set of distinct, interpretable motifs (Wiltschko et al., 2015). A family of dynamical system models aim to discover these discrete state transitions in an unsupervised manner. In particular, switching linear dynamical systems (SLDSs) formalize this observation by inferring a decomposition of the complex dynamics into locally linear dynamics primitives (Ackerson & Fu, 1970; Barber, 2006; Linderman et al., 2017; Glaser et al., 2020; Chen et al., 2024). Only one of the dynamics primitives is used to describe the underlying data at any time point, which is defined as the state of the system. The model learns to switch between states to improve accuracy, enabling interpretable explanations of the observations. However, many real-world dynamics display extended, soft, stochastic transitions between states. In such cases, interpretability of SLDS models diminishes. Moreover, switching between discrete states is prone to spurious rapid switching under the influence of complex noise processes across multiple states, a phenomenon commonly observed in real datasets.

More broadly, while desirable for interpretability, discreteness poses challenges in analyzing the physical world. One relevant manifestation is the difficulty of incorporating discrete factors into machine learning models: although gradient descent fuels spectacular successes, obtaining gradient estimates around such discrete factors is inherently problematic. The Gumbel distribution, a member of the extreme value distribution family (Gumbel, 1935; 1941), offers a relaxation to produce "soft discrete" samples, where the approximation is controlled by a temperature parameter (Jang et al., 2016; Maddison et al., 2016). Here, we adopt this approach to propose a dynamical model that approximates switching dynamics, is trained with gradient descent, and offers interpretable characterizations even when the parameter estimates deviate substantially.

The Gumbel-soft relaxation of states, the soft transition design of the dynamics, and the efficient inference algorithms together provide several advantages for analyzing complex time series. First, the model accommodates systems with mixed states and stochastic transitions. Second, the soft relaxation reduces spurious rapid switching, leading to more interpretable notions of states. Finally, the models are fast to train and generalize readily to unseen data. We validate our approach on benchmark simulations and two real-world datasets; Formula 1 race telemetry data (Schaefer, 2020) and the Caltech Mouse Social Interactions dataset (CalMS21) (Sun et al., 2021). We observe that our implementation learns faster and produces more interpretable state estimates compared to competitive benchmarks.

## 1.1 RELATED WORK

Our model is related to the family of state-space models, including autoregressive hidden Markov models (AR-HMMs) and switching linear dynamical systems (SLDSs). AR-HMMs extend standard HMMs by incorporating autoregressive observations, making them suitable for modeling nonlinear temporal dependencies in time series (Juang & Rabiner, 1985; Guan et al., 2016). The switching linear dynamical systems (SLDSs), first proposed by Ackerson & Fu (1970), decompose complex time series data into sequences of simpler linear dynamics primitives. Linderman et al. (2017) extended SLDSs to recurrent SLDSs (rSLDSs), allowing discrete state transitions to depend on the continuous latent state of the system or environment. Glaser et al. (2020) further extended rSLDSs to model interactions across multiple populations. Dong et al. (2020) studied the recurrent nonlinear SLDS (rSLNDS) and proposed a collapsed variational inference approach for efficient inference. Ansari et al. (2021) extended the rSLNDS framework by augmenting the nonlinear continuous dynamics with explicit-duration variables to model sojourn times for each discrete state. More recently, Hu et al. (2024) developed a framework that extends rSLDS by introducing a Gaussian Process prior that allows smooth state switches at the boundaries of linear dynamical regimes.

Recent studies have recognized the need for models that preserve interpretability while maintaining a high level of expressivity. A key idea is decomposing complex time series data into linear dynamical systems (LDSs). Fraccaro et al. (2017) proposed the Kalman VAE, which combines a variational auto-encoder with linear Gaussian state-space models and learns a separate dynamics-parameter network that captures the time-varying weighting of each linear Gaussian state-space model. Mudrik et al. (2024) decomposed transitions between consecutive time points as a time-varying mixture of LDSs. Chen et al. (2024) extended this to probabilistic decomposed linear dynamical systems (p-dLDS), introducing hierarchical random variables that encourage sparse and smooth dynamics coefficients. While p-dLDS improves dLDS on robustness to noise, it removes the notion of discrete states and their recurrent relationships with the environment. More recently, TiDHy, a hierarchical generative model proposed by Abe & Brunton (2025), learns to demix timescales by decomposing dynamical systems into simultaneous orthogonal LDSs operating at different timescales.

The use of the Gumbel-Softmax distribution as a differentiable sampling or reparametrization tool has been explored to varying degrees by prior works in the literature on switching linear dynamical systems. Fraccaro et al. (2017) first remarked in its appendix that the dynamics-parameter weights in KVAE could be approximated as discrete random variables using the Gumbel distribution. Becker-Ehmck et al. (2019) proposed a differentiable SLDS by replacing the categorical discrete states in SLDS with a Gumbel-Softmax relaxation to enable gradient flow. Moreover, the Gumbel-Softmax SNLDS proposed by Dong et al. (2020) as a baseline model uses Gumbel-Softmax relaxation in the variational posterior as a substitute for marginalizing over discrete states. We discuss a detailed comparison to these works in Appendix A, and we emphasize that the GDM we propose here is a dynamical system explicitly driven by Gumbel noise, rather than a soft mixture or an auxiliary inference trick.

GDM is not restricted to the classical SLDS parameterization and can incorporate expressive sequence models such as RNNs within its components. Modern architectures—including neural differential equations (Chen et al., 2018), S4 (Gu et al., 2021), transformers (Vaswani et al., 2017), and recurrent deep networks—can achieve remarkable performance in sequence prediction and function approximation. However, these models are generally not designed to produce switching latent dynamical primitives and temporal intervals. Combining Gumbel-driven state dynamics with such architectures, for instance by coupling attention mechanisms or continuous-time models with learn-

able state switching structure, offers a promising future direction that could unify the expressiveness of deep sequence models with the interpretability of discrete dynamical structure.

## 1.2 SUMMARY OF CONTRIBUTIONS

Our contributions can be summarized in the following points.

• We propose a new dynamical system model based on a Gumbel noise model defined over a relaxed-discrete state space. It infers interpretable states from complex time series with non-stationary, nonlinear dynamics.

• We define a differentiable variational posterior directly over states, enabling fast, scalable training with standard gradient descent methods. We optimize with respect to state dynamics end-to-end.

• We design an amortized inference network that parameterizes the variational posterior of the states. Fully amortized variational inference lets the model generalize immediately to unseen examples *without re-optimizing a per-sequence latent trajectory posterior*, in contrast to many existing methods.

• We evaluate performance using metrics that capture both fit and quality of the inferred states. Our model consistently outperforms competitive benchmarks and infers more interpretable state estimates on simulation and complicated real-life datasets.

## 2 MODEL FORMULATION

### 2.1 GUMBEL-SOFTMAX TRICK

The Gumbel–Softmax trick Jang et al. (2016); Maddison et al. (2016) provides a continuous relaxation of discrete random variables, enabling gradient-based optimization. Specifically, given logits $\pi \in \mathbb{R}^K$ corresponding to a categorical distribution, the trick proceeds as follows. Let $G(\mu, \beta)$ denote the Gumbel distribution with location $\mu$ and scale $\beta$ Gumbel (1941). We sample Gumbel noises $g_i \sim G(0, 1)$ and form perturbed logits $\pi_i + g_i$. The maximum $\max_i\{g_i + \pi_i\}$ follows a Gumbel distribution with location parameter $\log \sum_j \exp(\pi_j)$ and scale 1, and the index $i$ that maximizes $g_i + \log \pi_i$ follows the categorical distribution. This is known as the Gumbel-Max trick, i.e.,

$$P(i = \arg\max_j (g_j + \pi_j)) = \frac{\exp(\pi_i)}{\sum_j \exp(\pi_j)}$$

Noting that the Gumbel is a member of the extreme-valued distributions family, Gumbel noise amplifies differences among competing logits, effectively sharpening the winner-take-all behavior behind the Gumbel–Max trick. A continuous relaxation replaces the argmax with a tempered softmax, which means that we can reparametrize the original discrete $z$ by a Gumbel-Softmax (GS) distribution, $z \sim \mathrm{softmax}\left(\frac{\pi+g}{\tau}\right)$, where $\tau$ is a temperature controlling the softness of the distribution. As $\tau \to 0^+$, the softmax converges to the argmax function and the GS distribution converges to the original categorical distribution. Note that the Gumbel-Max trick is invariant to identical shifts in the location parameter $\mu$. On the other hand, the scale parameter $\beta$ controls the spread of the Gumbel noise added to logits. If we sample Gumbel noises $g$ from $G(0, \beta)$ instead of $G(0, 1)$, the effective softmax becomes $z \sim \mathrm{softmax}\left(\frac{\pi/\beta+g}{\tau/\beta}\right)$.

For simplicity, we fix the scale parameter $\beta = 1$ and denote this reparameterization as $z \sim \mathrm{GS}(\pi, \tau)$. In this way, we have differentiable $q(z|\phi)$ with continuous GS $z$ sampled from fixed, parameter-free Gumbel noises. In practice, we usually set the temperature $\tau$ to a moderate value to ensure smooth gradient flow in training. This also explicitly accounts for uncertainty in state transitions. Because of the extreme-value behavior of the Gumbel distribution, the resulting GS samples remain close to one-hot under moderate temperatures, preserving the semantics of discrete states while still enabling smooth optimization. Gumbel dynamical model, to be introduced in the next section, then leverages this heavy-tailed, winner-dominant behavior as a mechanism for modulating stickiness and competition among latent states, thereby preserving interpretable switching dynamics without enforcing hard discreteness. We leave more background details to Appendix B.

## 2.2 GUMBEL DYNAMICAL MODEL

We propose a new dynamic switching model to accommodate continuous Gumbel-Softmax state samples, the Gumbel Dynamical Model (GDM):

$$z_1 \sim \text{GS}(\pi_1, \tau), \qquad z_t \mid z_{t-1}, y_{t-1} \sim \text{GS}(\pi_t, \tau), \quad \pi_t = f_\theta(z_{t-1}, Fy_{t-1}), \quad t \geq 2, \quad (1)$$

$$y_1 \mid z_1 \sim \mathcal{N}(z_1 \cdot \mu, R), \qquad y_t \mid y_{t-1}, z_t \sim \mathcal{N}\big(\sum_k z_{t,k}(S_k Fy_{t-1} + b_k), R_t\big), \qquad t \geq 2.$$

Here, $\pi_1$ is a learnable prior over states, $\mu$ is an observation prior, $S_k \in \mathbb{R}^{N \times D}$ captures state-dependent dynamics in the projected observation space, $F \in \mathbb{R}^{D \times N}$ projects observations to a low-dimensional latent space, and $R_t$ models the observation covariance. Importantly, $f_\theta$ can be any feed-forward network parameterized by $\theta$. As a simple and interpretable case, $f_\theta$ can take a linear recurrent form $f_\theta(z_{t-1}, Fy_{t-1}) = RFy_{t-1} + r$, where $R$ is a learnable $K \times D$ transition matrix and $r$ is a bias vector. To explicitly encourage persistence, a sticky variant mixes the logits with the previous soft state: $\pi_t = (1 - \gamma)(RFy_{t-1} + r) + \gamma z_{t-1}$.

The Markov-1 assumption in the GDM can be relaxed to incorporate longer history. In this case, we parametrize the transition logits with an RNN: let $h_t$ be the hidden state updated as $h_t = g(h_{t-1}, Fy_{t-1})$ where $g$ is a recurrent architecture such as GRU. We then define the transition logits as $\pi_t = \text{FNN}(z_{t-1}, h_t)$. While the state dynamics become non-linear, the soft states $z_t$ still correspond to interpretable dynamical motifs, preserving the interpretability of the model. Unless otherwise stated, we refer to the GDM in its linear sticky form.

In GDM, the observation $y_t$ at time step $t$ feeds back into the state dynamics through the projection matrix $F$, such that $Fy_t$ recovers the low-dimensional latent trajectory. In fact, GDM can be related to the family of switching linear dynamical systems (SLDS) by introducing a latent projected observation $x_t = \mathbb{E}[Fy_t \mid z_{\leq t}]$ for $t \geq 1$, where the expectation is taken conditional on all past states. Note that this expectation removes the direct dependence of $z_t$ on $y_{t-1}$ for all time step $t$. Replacing $Fy_{t-1}$ in the GDM with $x_{t-1}$ yields a two-level GDM system, which is equivalent to

$$z_1 \sim \text{GS}(\pi_1, \tau), \quad z_t \mid z_{t-1}, x_{t-1} \sim \text{GS}(\pi_t, \tau), \quad \pi_t = f(z_{t-1}, x_{t-1}), \quad t \geq 2, \quad (2)$$

$$x_1 = z_1 \cdot \mu, \quad x_t \mid x_{t-1}, z_t = \sum_k z_{t,k}(A_k x_{t-1} + c_k), \quad t \geq 2,$$

$$y_t \mid x_t \sim \mathcal{N}(Cx_t, Q_t), \quad t \geq 1.$$

Here, the continuous latent trajectory $x_t$ at time $t$ is determined by a mixture of dynamics over the soft states $z_t$. Importantly, $x_t$ is deterministic given $z$, and is introduced to facilitate interpretation. At each time $t$, $x_t$ can be viewed as the expected projection of $y_t$. Uncertainty in the system is thus captured solely by the Gumbel noise on $z$ and the Gaussian noise on $y$. Figure 1 illustrates the graphical models of both systems, highlighting their relationships and dependencies. A proof of system equivalence is provided in Appendix C.

More generally, one could allow additional noise in the latent trajectory $x$ by introducing state-dependent covariances. This results in a mixture version of the standard recurrent SLDS with Gumbel state dynamics. Although more expressive in principle, the trajectory dynamics $x$ and the state dynamics $z$ compete to explain the data, and inference becomes more expensive as a flexible posterior is required to capture their intricate dependencies. For completeness, we discuss variational inference for this 3-level mixture model in Appendix D.

Finally, we note that this 3-level model is non-identifiable. In particular, the latent trajectory $x$ is only recoverable up to an affine transformation. For GDM, while the projection matrix $F$ and dynamic matrices $S_k$ are identifiable only up to an invertible linear transformation, the remaining parameters are identifiable up to permutations (Balsells-Rodas et al., 2023) in the limiting case $\tau \to 0$. While establishing a full identifiability theory for the non-limiting case is nontrivial, we note that the introduction of Gumbel noise does not create qualitatively new sources of non-identifiability. We provide a more detailed discussion of these points in Appendix E. Importantly, GDM improves state estimation by removing the trade-off between stochasticity in the continuous latent trajectory $x$ and stochasticity in the switching state $z$, thereby enhancing interpretability in practice.

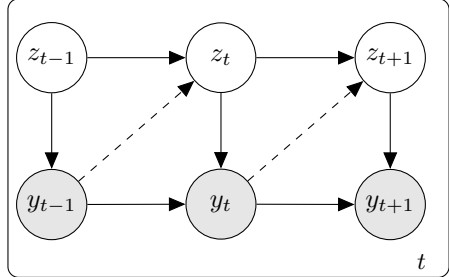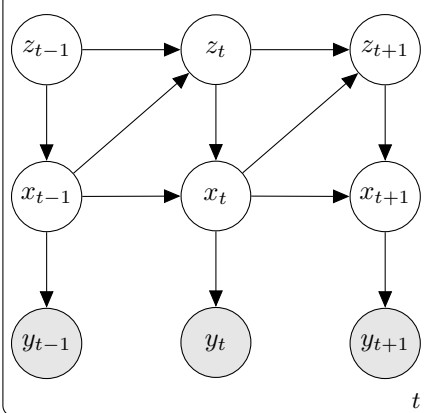

Figure 1: **Graphical model representation of two systems.** Left: 2-level GDM. Right: 3-level Mixture SLDS. Dashed lines denote dependencies that can be removed to make the two systems equivalent.

## 3 MODEL INFERENCE

Due to the continuous nature of states, GDM can be trained using standard gradient descent. To infer the GDM, we use BBVI (Ranganath et al., 2014) with Gumbel-Softmax samples (GS-BBVI): we define variational distribution $q(z)$, sample soft states $z$ from $q(z)$, and compute unbiased samples of the ELBO gradient.

**ELBO.** The ELBO for the GDM can be written as follows,

$$\log p_\theta(y_{1:T}) \geq \mathbb{E}_{q(z)} \log(y, z) - \log q(z)$$

$$= \mathbb{E}_{q(z)} \left[ \sum_{t=1}^{T} \log p(y_t|y_{t-1}, z_t) + \sum_{t=2}^{T} \log p(z_t|z_{t-1}) + \log p(z_1) \right] - \mathbb{E}_{q(z)} \left[ \log q(z_{1:T}) \right]$$

### 3.1 VARIATIONAL POSTERIORS

We approximate the posterior over latent states with an amortized variational distribution $q_\phi(z_{1:T} \mid y_{1:T})$, parameterized by a neural network that maps observations to Gumbel-Softmax logits. Specifically,

$$q_\phi(z_{1:T} \mid y_{1:T}) = \prod_{t=1}^{T} q_\phi(z_t \mid y_{1:T}),$$

where each $z_t$ is a continuous Gumbel-Softmax random variable with logits $\pi_t'$ and temperature $\tau$.

Since $z_1, \ldots, z_T$ are continuous Gumbel-Softmax random variables, we cannot directly define a discrete transition matrix as in the categorical case. Instead, we define a function that computes the logits $\pi_1', \ldots, \pi_T'$. Here, the logits $\pi_{1:T}'$ are produced by an inference network $g_\phi(y_{1:T})$ that shares a similar structure to the transition network $f_\theta$ in the generative model, i.e., $g_\phi$ may be a simple feed-forward mapping or a recurrent network. In principle, $g_\phi$ can be more expressive than $f_\theta$. This flexibility can improve posterior approximation and accelerate training. However, in practice, a highly expressive $g_\phi$ may compensate for the limitations of $f_\theta$, leading to posteriors that fit the observations well but provide less interpretable dynamics. For this reason, in this paper we keep the structures of $g_\phi$ and $f_\theta$ aligned.

Concretely, if $f_\theta$ is linear, $g_\phi$ can be chosen as a linear map, e.g., $\pi_t' = W y_t + b$. Optionally, a sticky component depending on $z_{t-1}$ can be introduced to encourage persistence, e.g., $\pi_t' = W y_t + B z_{t-1} + b$, with $z_1$ drawn from a Gumbel-Softmax distribution parameterized by learnable prior

logits $\pi'_1$. In this case, the variational posterior admits a Markovian factorization,

$$q(z_{1:T}|y_{1:T}) = q(z_1 \mid y_1) \prod_{t=2}^{T} q(z_t \mid z_{t-1}, y_t),$$

If $f_\theta$ is recurrent, we instead parameterize $g_\phi$ with a bidirectional RNN or a Transformer, so that $\pi'_t$ depends on both past and future observations. Temporal dependencies between observations are captured implicitly by the shared hidden states of the RNN. This yields a more expressive posterior that leverages temporal context to infer $z_t$. Concretely, for example, let $e_{1:T} = \text{BiGRU}(y_{1:T})$, and set $\pi'_t = \text{FNN}(z_{t-1}, e_t)$.

Thanks to the Gumbel-Softmax reparameterization trick, we can sample $q(z)$ sequentially in a differentiable way. The temperature $\tau$ for the Gumbel-Softmax distribution controls the smoothness of the state transition. Empirically, we find that GDM's behavior is largely invariant to the Gumbel–Softmax temperature over a broad range $\tau \in (0.5, 1)$. (We did not test $\tau \geq 1$. $\tau$ was either constant or followed a non-increasing schedule during training.) This is due to the extreme-valued nature of the Gumbel distribution as most samples cluster around the corners of the simplex. In practice, fixing $\tau \approx 1$ typically provides both stable optimization and interpretable state recovery. Incorporating an annealing schedule that starts at a higher temperature and gradually reducing to the target value can further improve robustness and flexibility. A relatively high temperature improves gradient-based optimization but produces less deterministic state boundaries. Therefore, accurate state recovery ultimately depends on learning the parameters that govern the latent state dynamics (e.g., the transition logits or their RNN parameterization), rather than relying on a low temperature alone to sharpen the state assignments.

Importantly, amortized variational inference with differentiable $q(z)$ is a key advantage of GDM. The inference network learns a reusable mapping from observations to state logits, enabling new data to be processed directly without re-optimization. This contrasts with many existing models, which typically require re-optimizing a posterior for the latent trajectory on each new dataset.

## 3.2 SMOOTHING AND PREDICTION

Once the variational posterior and model parameters are trained, the inferred system can be used for smoothing current observations, evaluating quality of fit, predicting future steps, and generating new observations.

Given a time series $y_1, \ldots, y_T$ of length $T$, we first obtain samples $z_1, \ldots, z_T$ from the variational posterior. Smoothed observations $\hat{y}_1, \ldots, \hat{y}_T$ are then computed based on the sampled states and past observations, providing a measure of reconstruction quality.

To predict future steps, we apply the learned transition model to generate next-step states $\hat{z}_2, \ldots, \hat{z}_T$ from the sampled states $z_1, \ldots, z_{T-1}$ and current observations $y_1, \ldots, y_T$. These predicted states are then used to generate corresponding next-step observations $\hat{y}_2, \ldots, \hat{y}_T$. The predicted observations can be recursively fed back into the transition model, enabling multi-step-ahead predictions. We note that an analogous procedure applies to the 3-level mixture formulation. Instead of propagating predicted observations, we propagate the inferred latent trajectory $\hat{x}_2, \ldots, \hat{x}_T$, which serves as input to the state transition function.

While this procedure can be extended to arbitrary horizons, uncertainty inevitably accumulates across steps. A $k$-step-ahead prediction for a series $y_1, \ldots, y_T$ is equivalent to producing $k$ future observations at each of the $T$ possible starting points. Because of the injected Gumbel noise in the latent states $z$, prediction trajectories may diverge after only a few steps, particularly at higher temperatures $\tau$. These divergent possibilities form a prediction envelope, whose width increases at points of greater transition uncertainty. This widening envelope corresponds naturally to the unpredictability observed in real-world dynamical systems. We will further illustrate this concept via simulation examples in section 4.

## 4 EXPERIMENTS

We validate the GDM on both simulated data and two real-world datasets. We begin with a standard, deterministic simulated example, then introduce soft, sticky, and stochastic transitions. We

further evaluate the model on two real-world datasets that feature multiple dynamic and highly unpredictable transitions. The code we use is available at: `https://anonymous.4open.science/r/GDM-CD3A/`.

To assess model performance, we use two metrics at different levels. At the observation level, we compute the coefficient of determination $R^2$ between the smoothed and true observations, which quantifies the quality of fit. At the state level, we introduce the following metric that measures the quality of inferred states.

**Inferred State Accuracy.** Let $\{\zeta_t\}_{t=1}^T$, $\zeta_t \in \{1, \ldots, K\}$, denote the ground-truth (or expert-labeled) discrete states, and let $\{z_t\}_{t=1}^T$, with $z_t \in \Delta^{K-1}$, denote the inferred states, where $\Delta^{K-1}$ is the $(K-1)$-simplex. In particular, discrete inferred states are represented as one-hot vectors in $\Delta^{K-1}$. We train a $k$-nearest neighbor (k-NN) classifier $f_{\text{KNN}} : \Delta^{K-1} \to \{1, \ldots, K\}$ on the training set by mapping inferred states $z_t$ to ground-truth $\zeta_t$. For test data $\mathcal{D}_{\text{test}}$, predictions are obtained as $\hat{\zeta}_t = f_{\text{KNN}}(z_t), \quad t \in \mathcal{D}_{\text{test}}$. The *Inferred State Accuracy* is then defined as

$$\text{Acc}_{\text{state}} = \frac{1}{|\mathcal{D}_{\text{test}}|} \sum_{t \in \mathcal{D}_{\text{test}}} \mathbf{1}\Big[\hat{\zeta}_t = \zeta_t\Big].$$

When the underlying ground truth $\zeta_t$ is manually obtained by human annotators, $\text{Acc}_{\text{state}}$ quantifies interpretability: it is high when it agrees with the human intuition and low otherwise.

### 4.1 FROM DETERMINISTIC TO UNCERTAIN: SYNTHETIC NASCAR DATASET

The synthetic NASCAR dataset (Linderman et al., 2017) emulates cars going around a track. It assumes four states in total: two for driving along the straightaways and two for the semicircular turns at each end of the track. The standard NASCAR setting assumes a nearly deterministic recurrent relationship between the current state and the previous trajectory. Since the states are determined by locations on the track, this construction yields a nearly fixed trajectory given the starting point. See Appendix F for construction details.

In this paper, we also consider a more realistic NASCAR trajectory that allows for soft state transitions and noise. This is achieved by replacing the recurrent relationship in Eqn. (8) with its soft sticky form:

$$z_t|x_{t-1} \sim \text{GS}(\pi_t, \tau), \text{s.t. } \pi_t = c(1-\gamma)(Sx_{t-1} + s) + \gamma z_{t-1} \quad t \geq 2 \tag{3}$$

where $c$ controls transition softness and $\gamma$ controls transition stickiness. As we decrease the scaling factor $c$, increase $\gamma$, and raise the temperature parameter $\tau$, GS samples become less deterministic and more noisy. Figure 2A shows qualitatively different trajectories from the same set of parameters.

We benchmark model performance against several models: SLDS with sticky transitions, rSLDS with sticky recurrent transitions, rSLDS with recurrent only transitions, p-dLDS, KVAE with MLP encoders, and SNLDS with collapsed variational inference. For both the standard and soft sticky NASCAR cases, we train models with four states (or dynamic operators) on the top trial and test on the bottom trial. All models achieve nearly perfect train $R^2$ on both datasets. For the soft-sticky case, however, all benchmark models except KVAE and SNLDS require retraining for variational posteriors to achieve good test $R^2$. Otherwise, the test $R^2$ is simply 0.8, i.e., the difference between the top and bottom trials. In contrast, our model achieves near-perfect test $R^2$ without retraining. This is because GDM employs amortized variational inference with differentiable variational posterior $q(z \mid y)$, as discussed in Section 3. We note that KVAE and SNLDS also generalize to test data without re-optimizing their variational parameters. However, KVAE achieves this by training two separate networks: a VAE that maps observations into a low-dimensional latent trajectory, and an additional dynamics-parameter network that maps this trajectory into time-varying dynamic weights. SNLDS, which inherits the collapsed variational inference technique, allows more expressive latent dynamics and uses amortized inference for the continuous latent variable. However, because the discrete switching variables are marginalized out rather than inferred directly, the discrete states must be recovered post-hoc, which limits their ability to capture interpretable states. For both cases, we repeated the training/testing procedure 10 times with different seeds.

Figure 2B shows the true and exemplar inferred states or dynamic weights from GDM, p-dLDS, KVAE and SNLDS. GDM successfully recovers the two dominant states in the soft sticky NASCAR

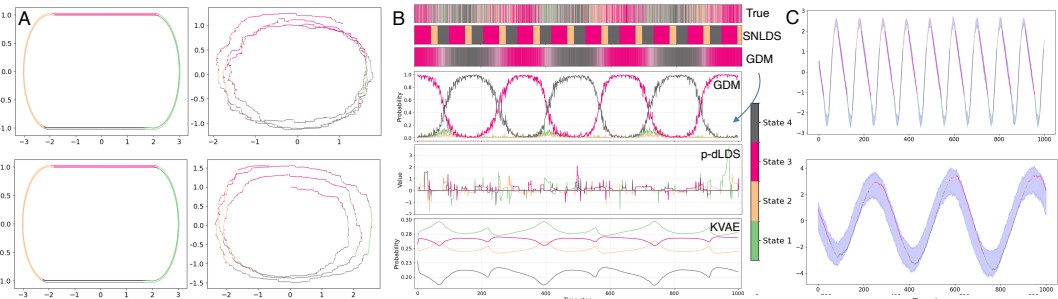

Figure 2: A. Standard and soft sticky NASCAR tracks. Two trials are generated with the same set of parameters ($T = 1000$, and $K = 4$ for both cases, $c = 0.02$ and $\gamma = 0.25$ for soft-sticky case only). Compared to the standard NASCAR, soft sticky NASCAR introduces greater transition uncertainty. B. True states and exemplar inferred states from GDM ($\tau = 0.99$), SNLDS, p-dLDS and KVAE. For each method, the panel shows the inferred state responsibilities over time: probabilities for GDM and KVAE, and scaler-valued dynamic coefficients for p-dLDS. Colored curves correspond to different dynamic primitives. Results shown are representative of 10 random seeds; full variability is reported in Table 1. C. Inferred 1-step-ahead prediction ranges for the first dimension of NASCAR observations. The top panel shows the standard model, and the bottom panel shows the soft sticky model, with a much wider uncertainty range. Shaded regions indicate ±3 standard deviations around the predicted mean, estimated from 100 Monte Carlo samples per model.

|    | SLDS (S) | rSLDS (S) | rSLDS (R) | p-dLDS | KVAE (M) | SNLDS (C) | GDM |
|----|----------|-----------|-----------|--------|----------|-----------|-----|
| S  | 0.82 ± 0.13 | 0.76 ± 0.10 | 0.96 ± 0.06 | 0.74 ± 0.01 | 0.67 ± 0.16 | 0.64 ± 0.02 | 0.88 ± 0.10 |
| SS | 0.32 ± 0.02 | 0.33 ± 0.01 | 0.43 ± 0.09 | 0.34 ± 0.02 | 0.50 ± 0.11 | 0.45 ± 0.05 | **0.70** ± 0.03 |

Table 1: Comparison of inferred state accuracy on the standard (S) and soft-sticky (SS) NASCAR datasets. "S" denotes sticky variants, "R" denote recurrent-only variants, "M" denotes the KVAE with MLP encoders, and "C" denotes the SNLDS with collapsed variational inference. Each model is trained and evaluated 10 times with different random seeds.

data, and approximates the other two states as combinations of dominant and complementary states. In contrast, all baseline models struggle to capture meaningful state structure in this setting. SLDS and rSLDS suffer from state collapse; p-dLDS utilizes all dynamic operators but fails to reproduce the correct oscillatory patterns; KVAE identifies the oscillations but yields noisy mixtures of dynamic weights, with the maximum state proportion at each time step remaining below 0.30; SNLDS also identifies the oscillation patterns but fails to capture the smooth transitions, overlapping states, and differences in the transition dynamics for the two dominant states. These limitations are consistent with known challenges of SDS-style models in regimes that depart from classical hard-switching assumptions, such as the soft-sticky settings we evaluate.

Table 1 reports the average state quality measured by mapping inferred states to hard-thresholded ground-truth states on the test trial. For the standard NASCAR data, rSLDS with recurrent only transitions achieves the top performance, while our model outperforms all the benchmarks in the soft sticky NASCAR case. Our model treats the observations as inherently stochastic, as discussed in section 3. While this uncertainty aspect is not advantageous in the standard NASCAR case, it allows the model to generalize better in the soft-sticky NASCAR case. Indeed, GDM correctly identifies that the soft sticky case exhibits greater uncertainty. This is illustrated by the one-step-ahead prediction envelopes in Figure 2C. While most one-step-ahead observations fall inside the envelopes for both cases, the envelope is clearly wider in the soft sticky case.

### 4.2 FROM SIMPLE STATES TO MORE STATES: F1 DATASET

The NASCAR dataset described above represents a simple track with four synthetic segments. Next, we consider a more complex and realistic example: the Formula One (F1) World Championship racetracks. A total of 77 circuits have hosted F1 races. Each F1 racetrack is uniquely designed

for its venue and is known for multiple challenging corners. We use the FastF1 package to retrieve telemetry data from past F1 sessions, including trajectory, lap times, and corner counts. In this paper, we study two permanent F1 circuits: the Shanghai International Circuit (China) and the Suzuka Circuit (Japan). For our purposes, we define track segments between consecutive numbered corners as distinct states. As shown in Figure 3A, the Chinese and Japanese Grands Prix have 16 and 18 corners, respectively. This definition of states is likely imperfect, but it is systematic and officially applied across all F1 circuits. We therefore expect that a good state representation should map to these expert-defined states with reasonable accuracy.

Since our model outperforms nearly all baseline methods except certain rSLDS variants in the synthetic NASCAR experiment, we benchmark GDM against rSLDS in this F1 dataset to further explore the model performance. As with NASCAR, we train models on one driver's trajectory and test on another's (Figure 3A). While drivers start from the same point, their speeds vary across laps, leading to trajectories of different lengths. For rSLDS, this requires retraining the variational posterior to infer latent states for a new driver. In this setup, both models achieve good training and testing fit.

However, rSLDS achieves good fit at the expense of state quality, particularly when the number of states $K$ is small. In other words, the optimizer improves likelihood at the cost of less interpretable states. To quantify this, we examine the state quality of both models for varying $K$ (Figure 3B). As shown in the plot, the state quality of the rSLDS is consistently lower than the GDM at all values of state dimension $K$. While rSLDS improves slowly as $K$ increases, GDM improves rapidly at the beginning steps and then sees a plateau. Although rSLDS may eventually reach reasonable inferred state accuracy for sufficiently large $K$, we note that smaller values of $K$ are usually preferred for interpretability in practice.

To illustrate interpretability concretely, we compare inferred trajectories for the Shanghai International Circuit at $K = 8$ (Figure 3C). GDM reveals four dominant states and approximates the remaining using combinations of available states. By contrast, rSLDS exhibits more frequent switching, failing to capture corner dynamics well in several cases.

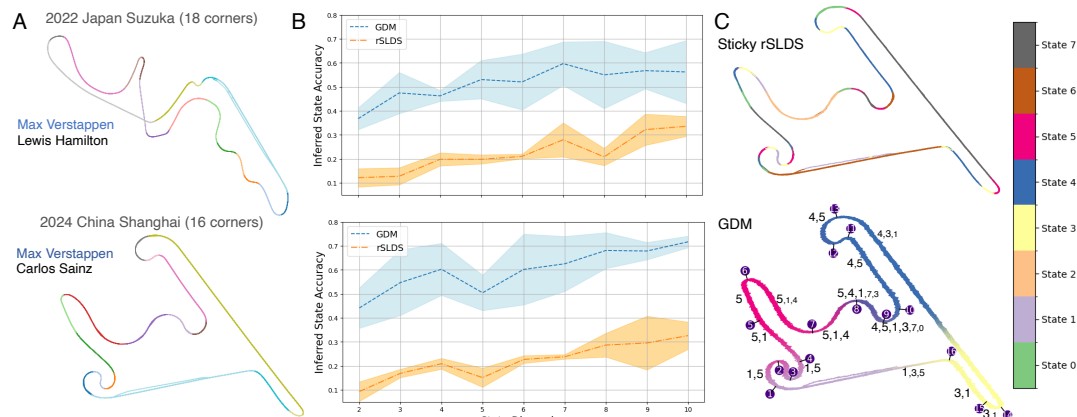

Figure 3: A. F1 Shanghai International Circuit (China) and Suzuka Circuit (Japan). Train trial: 1st-place winner (blue). Test trial: 5th-place finisher (black). B. Comparison of inferred state accuracy between our model and rSLDS across state dimensionalities. Performance is evaluated over 5 train/test splits with different random seeds. The shaded region denotes the standard deviation across seeds. GDM consistently achieves higher inferred state accuracy, particularly at low dimensions. C. Example inferred trajectories for both models on the Shanghai International Circuit. Results shown are representative of the 5-seed experiments. For GDM, we annotate each segment with state IDs that exceed 1% weight in at least 20% of the time steps associated with the corresponding expert-labeled segment. Note that the state IDs are ordered by presence ratio, and their marker sizes roughly reflect their weights. See Appendix G for further discussion of state usage.

### 4.3 UNCERTAINTY AND MULTIPLE STATES: CALMS21 DATASET

Finally, we apply our method to study mouse social behavior using the first task in the open CalMS21 dataset Sun et al. (2021). The dataset consists of location data for two mice interacting in a cage from multiple trials (89 trials, split into 70 train and 19 test) over 5 years. Each mouse is labeled with 7 keypoints, corresponding to the nose, ears, base of neck, hips, and tail (Figure 4A). As there are 14 keypoints with x, y values per frame, the observation dimension is 28. Importantly, this dataset is expert labeled. All frames in the 89 trials are manually labelled by one expert for four distinct social behaviors (attack, investigation, mount and other).

This dataset is a good candidate for our model, as the mouse behavior is highly unpredictable, and potentially includes multiple intricate states. We train our models on the 70 training trials, and test it on the 19 test trials, fixing the state dimension as $K = 5$.

Figure 4 summarizes the performance of our models (linear-sticky and RNN-based GDMs) alongside the rSLDS benchmark. Both GDM variants achieve higher training and testing accuracy than rSLDS for nearly all trials in this dataset. Consistent with our findings on the F1 dataset, our models also yield substantially higher inferred-state accuracy across all test trials. We demonstrate this via an exemplar training session, shown in Figure 4.

A key observation emerges when comparing the two GDM variants. The RNN-based GDM model achieves the highest observation-level accuracy for both training and test sets, reflecting the benefit of incorporating nonlinear recurrent functions into both the generative model and the variational posterior. However, its inferred-state accuracy is lower than that of the linear–sticky version. This underlines a key trade-off: adding expressive RNN/ bidirectional RNN components improves predictive accuracy but comes at the cost of decreased interpretability in the latent state dynamics.

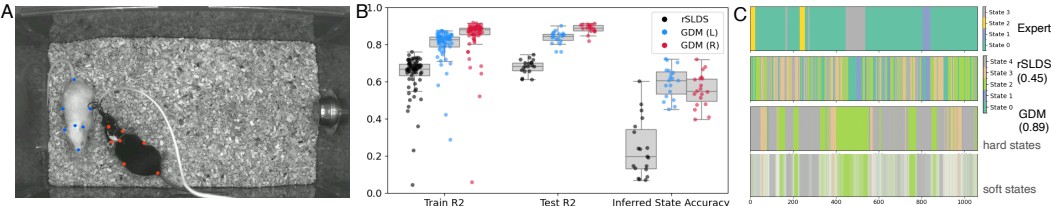

Figure 4: A. Example frame from the CalMS21 data. Seven anatomically defined keypoints are labeled on the body of each mouse. Expert annotations refer to behaviors initiated by the black mouse. B. Comparison of train $R^2$, test $R^2$, and inferred state accuracy between our models and the benchmark model rSLDS. "L" denotes the linear sticky GDM, and "R" denotes the RNN-based GDM. C. Expert-labeled states and inferred states from linear-sticky GDM and rSLDS, for trial 34 (the shortest trial containing all states). Accuracy values in brackets denote the inferred state accuracy with k-NN fitted directly on this trial. For details on state visualization, refer to Appendix G.

## 5 CONCLUSION

In this work, we proposed a dynamical system model to decompose complicated dynamics into simpler components that are referred to as states. We achieved this by relaxing the discreteness constraint on the states using the GS machinery. Therefore, our model breaks from previous work by using a latent dynamics noise model that is not Gaussian. The GS relaxation enabled us to model extended and soft transitions between states, identify states that may be implemented by a sparse combination of state primitives, and utilize the speed and ubiquity of standard gradient descent. We observed that this approach significantly improved the alignment of inferred states with available state annotations on complicated, real-world tasks. While GDM will benefit the analysis of dynamical systems on a wide range of topics, we think a better characterization of the impact of the Gumbel parameters on GDM's performance will be key to future improvements.

**Ethics statement**   We conceived GDM as a tool to improve analysis of dynamical phenomena. While we hope that it will benefit the society in the longer run by supporting progress across scientific disciplines, we do not think our work carries any immediate societal impact.

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

## A   COMPARISON TO GUMBEL-SOFTMAX LINE OF WORK

In this section, we provide detailed comparisons to the prior studies using Gumbel-Softmax in the literature of switching linear dynamical systems in terms of core modeling assumptions, dependency structure, the role of the Gumbel distribution, and inference compatibility. Notably, all works use Gumbel noise only as a differentiable sampling or reparameterization tool within the inference network and rely on specialized structured variational schemes, whereas GDM is a Gumbel-driven dynamical system whose latent evolution is directly governed by the Gumbel distribution and remains fully compatible with standard amortized BBVI.

**Comparison to KVAE**  GDM has a fundamentally different graphical structure from the KVAE proposed by Fraccaro et al. (2017). As illustrated in Figure 1 of both papers, KVAE combines a VAE with a linear Gaussian state space model (LGSSM): observations are first mapped into a low-dimensional latent space by a deep neural network, and those latents are then explained by a soft mixture of LGSSMs. The dynamics-parameter network in KVAE (Section 3.3) corresponds conceptually to the dynamic-operator weighting in p-dLDS (Chen et al., 2024), where the model learns continuous weights over multiple linear dynamical components.

Crucially, KVAE imposes no structural prior, sparsity constraint, or regularization on these mixture weights, so the learned dynamics tend to be dense mixtures, not interpretable switches. This limitation is directly visible in our experiments: using the publicly released KVAE code, we evaluated the model on the NASCAR benchmark. Table 1 shows consistently low discrete-state inference accuracy for KVAE compared to our model. Moreover, the Figure 2B illustrates an exemplar dynamics-parameter network weighting derived from KVAE, in which the maximum state weights are below 0.30 throughout the time span.

The KVAE appendix remarks that these weights could be "approximated as a discrete random variable using the Gumbel distribution," but this remark is not accompanied by any architectural change, dependency modification, or implementation. The paper does not specify how a discrete dynamics variable would interact with the KVAE structure or how such a model would be inferred.

**Comparison to relaxed SDLS**  Although GDM shares the use of the Gumbel distribution with relaxed SLDS proposed by Becker-Ehmck et al. (2019), the resulting generative model is fundamentally different. Becker-Ehmck et al. (2019) use the Gumbel relaxation as a gradient-flow tool for an otherwise standard SLDS; the Gumbel variables are not part of the generative process, whereas GDM introduces a Gumbel-driven dynamical system that cannot be interpreted as a relaxation of any standard SDS model. Importantly, GDM directly links states to observations and does not blur latent dynamics with Gaussian noises. In contrast, as the authors explicitly note below model formulation, "We do not condition the likelihood for the current observation directly on the switching variables", meaning the discrete variable only selects a transition dynamic for the continuous latent space. We further show that removing the important observation-to-state dependency makes GDM equivalent to a three-level mixture SLDS with a deterministic intermediate layer, which is introduced purely for interpretability. Comparing the graphical representations in Figure 1 of both papers highlights these structural differences.

We note that the Becker-Ehmck et al model is conceptually similar to the prototype we discussed in Appendix D . We explicitly analyzed its weaknesses, mainly, the continuous and discrete dynamics compete to explain the data; and proposed a BBVI-based solution for completeness. We also implemented this variant early in development but found it unsatisfactory — performing worse than GDM in both accuracy and speed, and scaling poorly to long, high-dimensional time series. Moreover, their inference procedure requires structured splitting and alternating updates, whereas our approach supports joint sampling and avoids customized inference machinery.

Finally, the modeling goals differ: in relaxed SLDS, the Gumbel variables are auxiliary and not evaluated for interpretability. Indeed, Section 5 reports that the Gaussian version performs comparably or better. Importantly, the authors did not report at all on the accuracy or interpretability of the inferred states. In contrast, we treat the Gumbel distribution as the driving noise of the dynamical system itself. GDM turns its heavy-tailed, extreme-value behavior into a way to modulate the stickiness and competition among states, leading to improved interpretability rather than only higher prediction accuracy.

**Comparison to GS-SNLDS**  The Gumbel–Softmax SNLDS of Dong et al. (2020) is technically conceptually and technically distinct from our work, despite superficial similarity. In their method, the Gumbel–Softmax relaxation appears only in the variational posterior as a stand-in for marginalizing discrete states. It does not define the switching dynamics, and the generative model remains a standard SNLDS. This makes their use of Gumbel comparable to Becker-Ehmck et al. (2019): the relaxation replaces the argmax inside the inference network only, not in the model. In contrast, GDM is a Gumbel-driven dynamical system in which Gumbel noise drives the switching process and determines the latent evolution. Moreover, while amortized inference is standard, the generative strucutre of GDM makes such amortization fundamentally simpler. In GDM, there is no need to

approximate $q(x)$ and no latent-to-latent stochasticity that gradients must pass through. Because observations depend directly on $z$, gradients propagate cleanly through the transition logits, making the model fully compatible with unmodified off-the-shelf amortized inference.

Additionally, we note that the GS-SNLDS variational posterior is structurally mismatched to its own generative model. It is mean-factorized so that the continuous latent posterior does not depend on the discrete state, even though in SNLDS the discrete state selects the transition dynamics. This breaks a core dependency in the model. This issue is reflected in their experimental results (Table 1): GS-SNLDS performs substantially worse than all other baselines including linear SLDS variants, despite SNLDS being strictly more expressive, empirically demonstrating that the proposed GDM is different in key aspects.

## B  BACKGROUND

**SLDS**   The standard SLDS model generates the observation $y$ from the continuous latent trajectory $x$ and the discrete latent state $z$. The discrete states $z \in \mathbb{R}^K$ can depend on the latent trajectory $x$,

$$z_t \sim \text{Cat}(\pi_t), \quad \pi_t = f(z_{t-1}, x_{t-1})$$

where $f$ can be linear or nonlinear. If the discrete state at time $t$ only depends on the latent trajectory at time $t-1$, the model is called recurrent only.

The continuous latent state $x_t \in \mathbb{R}^D$ follows conditionally linear dynamics determined by state $z_t$,

$$x_t \sim \mathcal{N}(A_{z_t} x_{t-1} + b_{z_t}, Q_{z_t})$$

where $A \in \mathbb{R}^{K \times D \times D}$ are the dynamics matrices, $b \in \mathbb{R}^{K \times D}$ are the shifts, and $Q \in \mathbb{R}^{K \times D \times D}$ are the covariance matrices. $K$ denotes the number of unique discrete states.

Finally, a linear Gaussian observation $y_t \in \mathbb{R}^N$ is generated from the corresponding latent state $x_t \in \mathbb{R}^D$,

$$y_t \sim \mathcal{N}(Cx_t + d, \sigma)$$

where $C \in \mathbb{R}^{N \times D}$ is the emission matrix. General stochastic optimization-based variational inference methods cannot be applied directly to SLDS due to the discreteness of the latent state $z$.

While the variational Laplace expectation-maximization (vLEM) algorithm is a popular choice for inference (Glaser et al., 2020; Zoltowski et al., 2020), it does not guarantee improvement in the evidence lower bound (ELBO) in the E-step because it relies on a second-order Taylor approximation around the mode of the posterior, which can be poor in high-dimensional or multimodal settings. On the other hand, general stochastic optimization-based variational inference methods like Black-Box Variational Inference (BBVI) cannot be applied directly to SLDS due to the discreteness of the latent state $z$.

**BBVI**   BBVI uses Monte Carlo gradients to optimize the ELBO. For an SLDS with latent variables $z, x$ and observation $y$,

$$\text{ELBO} = \mathbb{E}_{q(z)} \left( \log p(x, z) - \log q_\phi(z) \right) \le \log p_\theta(x)$$

To optimize the ELBO with stochastic optimization, consider the gradient of the ELBO as expectation with respect to the variational distribution,

$$\nabla_\phi \text{ELBO} = \mathbb{E}_{q(z,x)} \left[ \nabla_\phi \log q(z, x | \phi) \left( \log p(y, x, z) - \log q(z, x | \phi) \right) \right]$$

Noisy unbiased samples of the ELBO gradient can be computed using Monte Carlo samples from $q(z, x)$.

$$\nabla_\phi \text{ELBO} \approx \frac{1}{S} \sum_{s=1}^{S} \nabla_\phi \log q(x_s, z_s | \phi) \left( \log p(y, x_s, z_s) - \log q(x_s, z_s | \phi) \right)$$

Note that the score function and sampling algorithms depend only on the variational distribution, not the underlying model. With samples from the variational distribution, the only requirement is the computation of the log joint $\log p(y, x_s, z_s)$.

## C  PROOF OF SYSTEM EQUIVALENCE

In this section, we derive the equivalence relationship between the mixture model and the 2-level GDM. Recall we defined the dependency-removed 2-level GDM as follows,

$$z_1 \sim \mathrm{GS}(\pi_1, \tau), \quad z_t|z_{t-1} \sim \mathrm{GS}(\pi_t, \tau), \text{ s.t. } \pi_t = f(z_{t-1}, \mathbb{E}(Fy_{t-1}|z_{t-1_\leq}))), \quad t \geq 2 \quad (4)$$

$$y_1|z_1 \sim \mathcal{N}(\sum_k z_{1,k}\mu_k, R_t), \quad y_t|y_{t-1}, z_t \sim \mathcal{N}(\sum_k z_{t,k}(S_k Fy_{t-1} + b_k), R_t), \quad t \geq 2$$

And we defined the 3-level mixture model as follows (see eqn. (2)),

$$z_1 \sim \mathrm{GS}(\pi_1, \tau), \quad z_t|z_{t-1}, x_{t-1} \sim \mathrm{GS}(\pi_t, \tau), \text{ s.t. } \pi_t = f(z_{t-1}, x_{t-1}), \quad t \geq 2 \quad (5)$$

$$x_1 = \sum_k z_{1,k}\mu_k, \quad x_t|x_{t-1}, z_t = \sum_k z_{t,k}(A_k x_{t-1} + c_k), \quad t \geq 2 \quad (6)$$

$$y_t|x_t \sim \mathcal{N}(Cx_t, Q_t), \quad t \geq 1 \quad (7)$$

Firstly, we derive the 3-level mixture model (2) from system 4. The state transition equation of model (2) follows from a straightforward substitution. To obtain eqn.(6), we consider

$$\mathbb{E}_{y_t|z_{t_\leq}}(Fy_t|z_{t_\leq}) = F\mathbb{E}_{y_{t-1}|z_{t_\leq}}[\mathbb{E}_{y_t|y_{t-1}}(y_t|y_{t-1}, z_{t_\leq})]$$

Splitting time steps before $t$ into time steps before $t-1$ and time step $t$ we have,

$$\mathbb{E}_{y_t|z_{t-1_\leq}, z_t}(Fy_t|z_{t-1}, z_t) = \mathbb{E}_{y_{t-1}|z_{t-1_\leq}, z_t} \sum_k z_{t,k} F(S_k(Fy_{t-1}) + b_k)$$

$$= \sum_k z_{t,k} \mathbb{E}_{y_{t-1}|z_{t-1_\leq}, z_t}(FS_k(Fy_{t-1}) + Fb_k)$$

$$= \sum_k z_{t,k}(FS_k x_{t-1} + Fb_k)$$

The last line is derived from the definition $x_{t-1} = \mathbb{E}(Fy_{t-1}|z_{t-1_\leq})$ and the fact that $y_{t-1}$ and $z_t$ are conditionally independent given $z_{t-1}$. Conditioning on $x_{t-1}$ and $z_t$, $x_t = \mathbb{E}_{y_t|z_{t_\leq}}(Fy_t|z_{t_\leq})$ is equivalent to the LHS of eqn.(6), as $x_{t-1}$ is fully determined by states before time step $t-1$. The RHS of the equation above can be put into RHS of eqn.(6) by setting $A_k = FS_k$, and $c_k = Fb_k$. Finally, to obtain eqn.(7), we consider the mean and variance of $y_t$. If we set $C = F^\dagger$, we have $\mathbb{E}(y_t|x_t) = Cx_t$. To obtain the variance, we consider

$$Q_t = \mathrm{Var}(y_t|x_t) = \mathbb{E}\mathrm{Var}(y_t|y_{t-1}, x_t, z_t) + \mathrm{Var}\mathbb{E}(y_t|y_{t-1}, x_t, z_t)$$

$$= R_t + \mathrm{Var}(\sum_k z_{t,k}(S_k Fy_{t-1} + b_k))$$

We can remove the dependency on $x_t$ in both summation terms, since $x_t$ is fixed given $z_t$ and $z_{t-1}$, and $y_t$ is independent of $z_{t-1}$ given $z_t$. In practice, we can assume a diagonal covariance structure $R_t = \sigma I$.

Next, we show the reverse derivation from the mixture model to the GDM.

To obtain the Gumbel dynamics equation for the GDM, we consider

$$\mathbb{E}_{y_t|z_{t_\leq}}(Fy_t|z_{t_\leq}) = \mathbb{E}_{x_t|z_{t_\leq}} \mathbb{E}_{y_t|x_t, z_{t_\leq}}(Fy_t|x_t, z_{t_\leq}) = \mathbb{E}_{y_t|x_t}(Fy_t|x_t)$$

The inner expectation reduces to $\mathbb{E}_{y_t|x_t}(Fy_t|x_t)$ as $y_t$ is independent of $z_t$ given $x_t$. The outer expectation can be removed as $x_t$ is fully determined by states before time step $t$.

By eqn. (7), we know that

$$x_t = \mathbb{E}_{y_t|x_t}(Fy_t|x_t) = \mathbb{E}_{y_t|z_{t_\leq}}(Fy_t|z_{t_\leq})$$

where $F = C^\dagger$. This gives the Gumbel dynamics equation for the GDM by substituting $\mathbb{E}(Fy_{t-1}|z_{t-1_\leq})$ in eqn. (5).

To derive the observation level for the GDM, we substitute eqn. (6) into eqn. (7). Specifically, we write $y_t = Cx_t + \epsilon$ where $\epsilon \sim \mathrm{N}(0, Q)$. Then we have,

$$y_t = C \sum_k z_{t,k}(A_k x_{t-1} + c_k) + \epsilon$$

$$= C \sum_k z_{t,k}(A_k(Fy_{t-1} - \tilde{\epsilon}) + c_k) + \epsilon$$

$$= \sum_k z_{t,k}(CA_k Fy_{t-1} + Cc_k) + \epsilon - \sum_k z_{t,k} CA_k \tilde{\epsilon}$$

where $\tilde{\epsilon} \sim \mathcal{N}(0, FQF^\mathsf{T})$ is another Gaussian noise term. The second line comes from eqn. (7), as we have $Fy_{t-1} = x_{t-1} + \tilde{\epsilon}$ where $\tilde{\epsilon} \sim \mathcal{N}(0, FQF^\mathsf{T})$. Therefore, if we set $S_k = CA_k$, $b_k = Cc_k$ and $R_t = Q + \sum_k z_{t,k} CA_k FQF^\mathsf{T} A_k^\mathsf{T} C^\mathsf{T}$, we recover the observation dynamics in GDM. Note that in the case that $Q$ is diagonal, $R$ is still a dense covariance matrix.

# D  VARIATIONAL INFERENCE FOR 3-LEVEL MIXTURE MDOEL

As discussed in the main text, inference for the general 3-level mixture model is more challenging as we need to define variational distributions for both the latent variables $x$ and $z$. We can define a flexible variational distribution $q(x, z)$ that allows dependency between $x$ and $z$. For $z$, we define the same form of variational posterior as above, with dependency on $x$ instead of $y$, i.e., $q(z_{1:T}) = q(z_1) \prod_{t=2}^T q(z_t | z_{t-1}, x_{t-1})$. For $x$, we introduce dependencies that span multiple time steps by assuming a Gaussian with block tri-diagonal precision for $x_{1:T}$.

$$q(x_{1:T}) = \mathcal{N}(x_{1:T} | \mu, \Sigma) = \mathcal{N}(x_{1:T} | J, h)$$

where $J$ is the precision matrix $J$ and $h$ is the linear potential, $\mu = J^{-1}h$ is the mean, $\Sigma = J^{-1}$ is the inverse precision (covariance) matrix. It can be written as the following pairwise linear Gaussian dynamics,

$$q(x_{1:T}) = [\prod_{t=1}^{T-1} \mathcal{N}(x_{t+1} | A_t x_t + b_t, Q_t)] \cdot [\prod_{t=1}^{T} \mathcal{N}(x_t | m_t, R_t)]$$

Note that it is easier to work with the pairwise LDS structure as the precision matrix $J$ can be efficiently inverted and sampled from. We assume that the transition parameters $A_t$, $Q_t$, and $b_t$ are state-dependent, $A_t = A_{z_t}$, $b_t = b_{z_t}$, and $Q_t = Q_{z_t}$.

**Sampling mechanism**  Note that sequential sampling is feasible for $z$ but not for $x$. Recall the standard way of sampling from $\mathcal{N}(\mu, \Sigma)$ as follows. If $\Sigma$ has Cholesky decomposition $\Sigma = LL^\mathsf{T}$, then we can generate samples using $x = \mu + L\eta$ where $\eta \sim \mathcal{N}(0, I)$. In our case, we need $z_t$ for all time steps $t$ to compute linear potential $h$ and inverse precision matrix $J$. To sample from $J$, we solve two equations: $J\mu = h$ and $U^\mathsf{T}\tilde{x} = \eta$ where $U$ is the Cholesky decomposition of $J$ s.t. $J = UU^\mathsf{T}$. The final sample of $x$ is the sum of $\mu$ and $\tilde{x}$.

To sample from $q(x, z)$, we first initialize the samples for $x$ using observation $y$. Then we sample from $q(z)$ sequentially as follows: 1) Sample $z_1$ from the GS distribution with $\phi_1$ 2) Compute logits $\phi_t$ using the learnable transition function and sample $z_t$ using the GS trick, for all $t \geq 2$. Based on samples for $z$, we continue sampling from $q(x)$ as described above.

**Complete ELBO**  The ELBO for the 3-level mixture model is:

$$\log p_\theta(\boldsymbol{y}_{1:T}) \geq \mathbb{E}_{q(x,z)} \log(y, x, z) - \log q(x, z)$$

$$= \mathbb{E}_{q(x,z)} \left[ \sum_{t=1}^{T} \log p(y_t | x_t) + \sum_{t=1}^{T} \log p(x_t | x_{t-1}, z_t) + \log p(z_1) + \sum_{t=2}^{T} \log p(z_t | z_{t-1}) \right]$$

$$- \mathbb{E}_{q(x,z)} \left[ \log q(x_{1:T} | z_{1:T}) + \log q(z_1) + \sum_{t=2}^{T} \log q(z_t | z_{t-1}, x_{t-1}) \right]$$

# E   IDENTIFIABILITY CONSIDERATIONS FOR GDM

In the limiting case $\tau \to 0$, the GDM has an equivalent formulation as a finite mixture model analogous to an AR-HMM. Following the same notations in Balsells-Rodas et al. (2023), for finite horizon $T$, one can define a bijective path indexing function $\psi$ that maps each $i \in \{1, \ldots, K^T\}$ to a set of states $z_{1:T}$. Then, the family of GDMs can be seen as a finite mixture over all possible discrete state paths.

Let $M_k = S_k F$ denote the product of the dynamic matrix $S_k$ and projection matrix $F$. At the observation level, GDM then satisfies the *unique-indexing assumption* on Gaussian means and initial states used in Balsells-Rodas et al. (2023). By Theorem 3.2 in their paper, under these conditions, the family of GDMs is identifiable up to permutations. Importantly, this identifiability result does not require restricting the form of the state transitions, and arbitrary recurrence from the switches is allowed Balsells-Rodas et al. (2025). In the case that transition logits $\pi_t$ depend only on the previous discrete state $z_{t-1}$, one can uniquely recover the transition matrix.

For $\tau > 0$, identifiability becomes more subtle. The continuous GS relaxation means that latent state $z_t$ at each time step $t$ takes values on the simplex $\Delta^{K-1}$, so GDM is no longer a finite mixture but rather behaves like an infinite mixture over continuous paths. The considerations above rely heavily on the use of finite mixture modeling techniques, and characterizing identifiability in the non-limiting regime remains an open problem. Nevertheless, we note that the introduction of Gumbel noise does not create qualitatively new sources of non-identifiability relative to the Gaussian noise injected into continuous latents in SDS models: the fundamental issues arise from symmetry classes and model over-specification, not from the specific choice of noise distribution. This provides intuition for how identifiability theory may extend to the non-limiting regime.

Developing a full identifiability theory for the non-limiting case will require new mathematical statements. A potential route toward a formal proof may draw on the ideas in Barndorff-Nielsen (1965).

# F   MORE DISCUSSIONS ON THE NASCAR DATASET

The full generative model used to simulate the NASCAR dataset is described as follows,

$$z_1 \sim \text{GS}(\pi_1, \tau), \quad z_t | x_{t-1} \sim \text{GS}(T x_{t-1} + t, \tau) \quad t \geq 2 \tag{8}$$

$$x_1 = \sum_{k=1}^{4} z_{1,k} \mu_k, \quad x_t | x_{t-1}, z_t = \sum_{k=1}^{4} z_{t,k} (A_k x_{t-1} + c_k) \quad t \geq 2 \tag{9}$$

$$y_t | x_t \sim \mathcal{N}(C x_t, \sigma I), \quad t \geq 1 \tag{10}$$

This can be achieved by setting extreme Gumbel-Softmax logits in eqn. (8). As an example, the transition matrix $T$ and the bias $t$ can be defined as

$$T = \begin{bmatrix} 10 & 0 \\ -10 & 0 \\ 0 & 10 \\ 0 & -10 \end{bmatrix} \qquad \mathbf{t} = \begin{bmatrix} -20 \\ -20 \\ -10 \\ -10 \end{bmatrix}$$

Eqn. (8) can be viewed as a classifier that divides the space into four regions such that the logit of each region $k$ is computed as $T_k \cdot x + t_k$ where $x \in \mathbb{R}$ denotes the point on the 2D trajectory. For example, if $x_1 > 2$ and $-1 < x_2 < 1$, the first logit will be greater than $0$ while other logits will be smaller than $0$, so the point is highly likely to be classified in state $k = 1$.

Eqn. (9) specifies how the system moves in each state. For the standard NASCAR, the ground truth dynamics matrices are defined as,

$$A_1 = A_2 = \text{expm}\left(\begin{bmatrix} 0 & \frac{\pi}{24} \\ -\frac{\pi}{24} & 0 \end{bmatrix}\right), \quad A_3 = A_4 = I = \begin{bmatrix} 1 & 0 \\ 0 & 1 \end{bmatrix}$$

where the first two states correspond to the semicircular turns of $7.5°$ at the end of the straight track. The ground-truth offsets are defined as,

$$c_k = \begin{cases} -(A_1 - I) \cdot \text{FP}_1, & k = 1 \\ -(A_2 - I) \cdot \text{FP}_2, & k = 2 \\ [0.1 \quad 0], & k = 3 \\ [-0.25 \quad 0], & k = 4 \end{cases}$$

where $b_1$ and $b_2$ specify rotations around $\mathrm{FP}_1 = (2, 0)$ and $\mathrm{FP}_2 = (-2, 0)$ at the semicircular turns, while $b_3$ and $b_4$ specify the constant speed along the straight track.

To model variable-speed transitions, we may introduce another parameter $s$ that denotes a varied speed for the dynamics equation (9) such that $\tilde{c}_k = sc_k$ where $s \in [s_{min}, 1]$ is uniformly sampled between a minimum low speed $s_{min}$ and full speed and is applied throughout each segment of the track. The observation is generated in the same way as before. Given the previous location in the trajectory $x_{t-1}$ and the current state $z_t$, we can generate the next trajectory point using eqn. (9). The trajectory is then mapped to the observations. Note that the shape of the trajectory will not be changed fundamentally by varying speed as the movement direction of each state remains unchanged.

## G    STATE USAGE AND VISUALIZATION

As mentioned in the main text, GDM utilizes all states, but not equally. In Figure 5A, we show the complete state usage of GDM for the trial illustrated in Figure 3C. For demonstration purposes, we display the first three laps around the track. As seen in the plot, while all states capture the three laps as three clear peaks in probability, States 1, 3, 4, and 5 are more dominant than the other four states. This is also reflected in the state annotations in Figure 3C. Here, we provide a more detailed version of Figure 3C by lowering the presence threshold to 5% of all time steps associated with the expert-labeled state. The complementary states for each segment are greyed out.

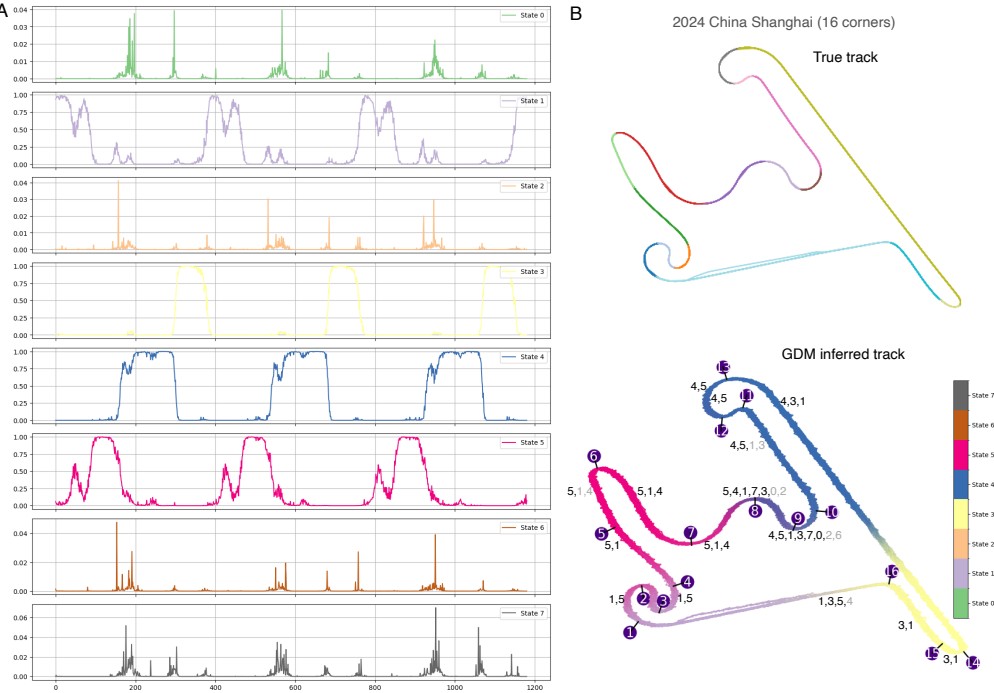

Figure 5: A. Complete state usages for Figure 3 B. Example inferred trajectory for GDM, with complementary states annotated in grey.

The unequal usage of states helps explain the observation that the inferred state accuracy of GDM improves rapidly in the initial steps and then plateaus. GDM allocates additional states to less dominant roles, so the marginal gain of increasing the number of states decreases after the first few.

For practical visualization, we put an emphasis on the dominant states. Specifically, we set transparency to the maximum value of state proportions at each time step and mix colors according to

the proportions of active states. This yields a gradual change in color across transitions and more transparent segments where mixtures of overlapping states occur.

## H  LLM USAGE

Large Language Models (LLMs) were used to assist with writing and polishing the manuscript and to improve the clarity and organization of the accompanying code repository.

