# OpenReview forum: "Interpretable time series analysis with Gumbel dynamics"
_ICLR.cc/2026/Conference — Submitted to ICLR 2026_

### Official Review · Reviewer_PpA9 · 2025-10-28

**Soundness:** 3
**Presentation:** 3
**Contribution:** 3
**Rating:** 6
**Confidence:** 3

**Summary:**

The paper proposes the Gumbel Dynamical Model (GDM), a switching dynamical system in which the discrete state is replaced by a Gumbel–Softmax continuous relaxation, enabling gradient-based training and “soft, sticky” state transitions that better capture variable-speed and overlapping regimes than classical methods. The generative model mixes per-state linear dynamics in a projected space to produce observations and uses amortized variational inference with a differentiable posterior over states. The paper shows an equivalence to a three-level mixture SLDS, discusses identifiability caveats, and argues that keeping noise only at the observation level avoids competition between state and latent-trajectory noise. Empirically, on synthetic NASCAR the method attains test “inferred state accuracy” of 0.88±0.10 in the standard setting and 0.70±0.03 in a harder “soft-sticky” setting where all baselines degrade to ≤0.43±0.09.

**Strengths:**

1. In terms of originality, relaxing the discrete state with a Gumbel-Softmax and designing the generative and inference maps to remain interpretable yields an expressive yet still human-legible model. The equivalence to a mixture SLDS clarifies what is novel versus inherited.
2. In terms of clarity, the model equations, training objective, and forecasting procedure are explained coherently with concrete choices, and the evaluation metric for state quality is operationalized via a k-NN trained on soft states to ground-truth labels.
3. In terms of significance, the combination of interpretable state mixtures, fewer rapid switches, and amortized inference that generalizes across trials makes GDM a practical tool for scientific time series with overlapping regimes where classic SLDS methods often fail.

**Weaknesses:**

1. The empirical study under-reports statistical variability, omits computational accounting, and leaves several core hyperparameters unexplored, such as the temperature, stickiness, softness, and projection dimension. These matters because the relaxation can blur state boundaries and the stickiness term changes dwell-time statistics.
2. Although the paper notes identifiability issues in the three-level formulation, it does not probe identifiability between the state-mixture dynamics and observation noise in the two-level GDM, nor does it quantify how increasing $\tau$ affects spurious blending versus accuracy.

**Questions:**

1. Please provide confidence intervals and seed counts for the results presented in tables and figures.
2. Please consider adding ablations regarding the terms mentioned in Weakness - 1.
3. How F is chosen for each dataset?

---

> ### Author Response · Authors · 2025-11-22
> **Thanks for the valuable technical suggestions!**
>
> We thank the reviewer for the positive assessment of the technical soundness, presentation, and novelty of our work, as well as for the constructive technical suggestions. We have addressed these points in detail below. Please find additional clarifications and updates included in the revised main text and appendix.
>
> **P1** *The empirical study under-reports statistical variability, omits computational accounting, and leaves several core hyperparameters unexplored, such as the temperature, stickiness, softness, and projection dimension. These matters because the relaxation can blur state boundaries and the stickiness term changes dwell-time statistics. Please provide confidence intervals and seed counts for the results presented in tables and figures. Please consider adding ablations regarding these terms. How $F$ is chosen for each dataset?*
>
> Thank you for the helpful suggestions. We have now included confidence intervals and seed counts for all experiments reported in tables and figures. Regarding the hyperparameters: except for the Gumbel–Softmax temperature (fixed at 0.99 in the initial submission), all other parameters are either learned directly during training (e.g., stickiness parameters and the projection matrix $F$) or set according to standard preprocessing schemes (e.g., PCA for dimensionality reduction). The softness is introduced and fixed for the soft-sticky NASCAR data generation process only; in our model, softness is not manually set but is instead learned through the state transition parameters. The fixed temperature $\tau=0.99$ was chosen because it is sufficiently low to yield near-discrete transitions while still maintaining stable optimization in PyTorch. Empirically, we found that GDM’s performance is largely insensitive to the exact temperature within a broad range, approximately $\tau\in (0,1)$. As suggested by Reviewer wAs1, we have also added an optional annealing scheme that starts training at a higher temperature and gradually reduces to the target value, which further increases training robustness and flexibility.
>
> **P2** *Although the paper notes identifiability issues in the three-level formulation, it does not probe identifiability between the state-mixture dynamics and observation noise in the two-level GDM, nor does it quantify how increasing in $\tau$ affects spurious blending versus accuracy.*
>
> Thanks for bringing up the identifiability consideration for the GDM. GDM improves identifiability relative to the three-level mixture model because there is no trade-off between stochasticity in the continuous latent trajectory $x$ and stochasticity in the switching state $z$. Following a recent work by Balsells-Rodas et al. (2024), we discuss the identifiability of GDM in the limit $\tau \to 0$ (see Appendix E).
>
>
> Regarding your question on how the Gumbel temperature affects state blending versus accuracy, we want to emphasize the extreme-value nature of the Gumbel distribution. This contributes to a relative insensitivity to temperature, as most samples remain very close to the vertices of the simplex for moderate $\tau$. Our experiments focus on interpretable state recovery, and we either fix or anneal the Gumbel temperature so that the model reaches a reasonably low temperature by the end of training. This setting produces near-discrete transitions and reliable state inference across datasets. While higher temperatures would in principle soften state assignments and could affect predictive accuracy, we do not systematically explore this regime because training is intended to conclude in the near-discrete regime. Our primary goal is to ensure stable and interpretable latent states, which is achieved under the chosen temperature.

---

> > ### Comment · Reviewer_PpA9 · 2025-11-27
> >
> > I thank the authors for the detailed rebuttal and the revised manuscript.
> > The authors have addressed my concerns. The empirical reporting is now more complete, and the identifiability and temperature behavior of GDM are better justified and documented. I still see room for future, more systematic temperature / dimension ablations, but I do not view this as a blocker. I therefore maintain my positive recommendation.

---

> > > ### Author Response · Authors · 2025-11-30
> > >
> > > Thank you again for your valuable technical suggestions and your supportive engagement throughout the review process. Your feedback has substantially improved the clarity and completeness of our paper, and we greatly appreciate it!

---

### Official Review · Reviewer_aNh3 · 2025-10-30

**Soundness:** 4
**Presentation:** 3
**Contribution:** 4
**Rating:** 8
**Confidence:** 3

**Summary:**

The paper introduces the Gumbel Dynamical Model (GDM), a differentiable and interpretable time-series model that extends dynamical systems. The key idea is to replace discrete latent states with continuous Gumbel-Softmax relaxations, enabling end-to-end gradient-based learning while preserving the interpretability of local dynamics. An amortized inference network predicts per-step logits for the Gumbel-Softmax states, forming a fully differentiable variational inference scheme.

**Strengths:**

1. The paper bridges discrete switching dynamical systems and differentiable deep models through a novel use of Gumbel-Softmax relaxation.
2. Derivations of Gumbel dynamics and the corresponding amortized variational inference are rigorous and internally consistent.
3. Extensive experiments are conducted on several datasets with different levels of complexity.

**Weaknesses:**

1. The model’s behavior seems to critically depend on the Gumbel-Softmax temperature, yet no sensitivity analysis is provided.
2. The “interpretable” property originates from the switching dynamical system itself, GDM mainly preserves this structure while making inference differentiable. A clearer formal definition or quantitative measure of interpretability would strengthen the claim.

**Questions:**

1. How sensitive are results to the temperature value?
2. Could GDM scale to high-dimensional observations?

---

> ### Author Response · Authors · 2025-11-22
> **Thanks for the detailed comments!**
>
> We thank the reviewer for the positive assessment of the technical soundness, presentation, and novelty of our work. We would like to provide further clarification on several technical aspects of our model. Please find our point-by-point replies below. Additional updates are included in our revised manuscript.
>
> **P1** *The model’s behavior seems to critically depend on the Gumbel-Softmax temperature, yet no sensitivity analysis is provided. How sensitive are results to the temperature value?*
>
> We thank the reviewer for raising this point. We have empirically found that GDM’s behavior is largely invariant to the Gumbel-Softmax temperature within a broad range $\tau\in(0.5,1)$, provided training concludes at a reasonably low temperature (e.g., $\tau \approx 1$). This is due to the extreme-valued nature of the Gumbel distribution as most samples cluster around the corners of the simplex. In the initial submission, we fixed $\tau=0.99$ which we found to be a practical choice ensuring both stable optimization and interpretable state recovery. We agree that an annealing schedule can provide additional robustness and flexibility; following suggestions from Reviewer wAs1, we have added an optional annealing mechanism that allows training to start at a higher temperature and gradually reduce to the target value near 1.
>
> **P2** *The "interpretable” property originates from the switching dynamical system itself, GDM mainly preserves this structure while making inference differentiable. A clearer formal definition or quantitative measure of interpretability would strengthen the claim.*
>
> We thank the reviewer for this insightful comment and would like to further emphasize the key novelty of GDM. Unlike prior Gumbel-Softmax approaches, such as Becker-Ehmck et al. (2019), which use Gumbel noise purely as a reparameterization trick for categorical sampling, GDM leverages Gumbel as a mechanistic dynamical prior that actively governs the evolution of the system among discrete primitives. The Gumbel distribution drives the latent switching process itself, and its heavy-tail, extreme-value behavior is used to modulate stickiness and competition between states. Structurally, GDM differs from the SLDS family in that there is no Gaussian-driven noise in the latent trajectory attempting to explain observations, and there is a direct dependency between the state dynamics and the observation. To provide a quantitative measure of interpretability, we define an inferred-state accuracy metric at the beginning of Section 4. This metric computes a state-matching score between inferred and true latent states, offering a simple yet meaningful quantification of interpretability in switching dynamical systems. To address the reviewer's concern, we have discussed this point in the revision while introducing the inferred-state accuracy.
>
> **P3** *Could GDM scale to high-dimensional observations?*
>
> Thank you for this question. Yes, GDM is designed to scale to high-dimensional observations while preserving interpretable dynamics and leveraging scalable amortized inference. Unlike prior works that rely on customized or structured variational inference, GDM requires no model-specific inference derivations, making it fully compatible with off-the-shelf amortized BBVI. For example, the number of observations $N$ in our simulated NASCAR example can easily be increased from 10 to 100 without a significant increase in training time, as the model is fully parallelizable and efficiently trained on GPU hardware.

---

### Official Review · Reviewer_gbmR · 2025-10-31

**Soundness:** 2
**Presentation:** 3
**Contribution:** 1
**Rating:** 2
**Confidence:** 5

**Summary:**

This paper proposes GDM, an estimation method for recurrent SLDSs and AR-HMMs. The methodology is based on introducing Gumble-softmax relaxations on the discrete variables. The latent variables are inferred via variational inference with MC sampling from the discrete posterior, and the model enables backpropagation thanks to the proposed relaxation. The paper demonstrates performance of GDM via synthetic NASCAR trajectories, F1 data, and mouse trajectories.

Unfortunately, the paper ignores critical related works which have specifically studies similar continuous relaxations of the discrete states for Switching Dynamical Systems.

**Strengths:**

- The presentation of the paper is adequate.
- The paper presents applications to real-world scenarios which are very promising for SDSs.

**Weaknesses:**

- The main concern is the novelty front of this paper. Unfortunately, GDM does not seem a novel methodology. See below a comparison of related works which might be relevant.
- The paper does not cite important related works that address inference in recurrent switching dynamical systems after Linderman's 2017 paper (sticky priors). There is a citation on a 2021 Neurips paper, but it involves group dynamics in SDSs. See below a list of critical related works:
  - Fraccaro et al. (2017) [1]: Kalman VAE. A Kalman filter in latent space, with a soft-switching mechanism. In their paper, they propose using a gumble-softmax sampling approach to estimate switches.
  - Becker-Ehmck et al. (2019) [2]: I believe this paper proposes exactly GDM (see Section 2.3 in their paper), for exactly the same type of SDSs (recurrent and linear).
  - Dong et al. (2020) [3]: This paper uses recurrent nonlinear SDS, it proposes exact inference on the switches "collapsed", but in the experiments they also propose an additional method as baselines which uses Gumble-softmax relaxation in a similar way as GDM. See Appendix A.2 in their paper. Importantly, their amortized inference network uses a bidirectional RNN to encode the sequence and a forward RNN to produce per-time-step switch posteriors, which matches the inference architecture proposed in this paper for GDM.
  - Ansari et al. (2021) [4]: Same as Dong et al. (2020), but the switches are explicit duration models.
- For acceptance, the paper should:
  - compare empirically to [1]–[4] on the proposed benchmarks, and
  - clearly articulate what is substantively new beyond those works. At present, I am not able to identify a principled difference between Becker-Ehmck et al. (2019) [2] and the proposed GDM; the submission presents GDM as novel, but it looks like a restatement of [2] (and of Dong et al.’s amortized inference design).
- The presented methodology lacks theoretical analysis on the presented methodology. Analysis on whether GDM is identifiable would be an important problem (i.e. the ground-truth representation is unique up to certain equivalence, e.g. permutaion of the switches). Without at least a statement of the assumptions under which GDM recovers meaningful modes, it is difficult to argue that the discovered “states” are interpretable rather than arbitrary mixtures.


[1] Fraccaro, Marco, et al. "A disentangled recognition and nonlinear dynamics model for unsupervised learning." Advances in neural information processing systems 30 (2017).

[2] Becker-Ehmck, Philip, Jan Peters, and Patrick Van Der Smagt. "Switching linear dynamics for variational bayes filtering." International conference on machine learning. PMLR, 2019.

[3] Dong, Zhe, et al. "Collapsed amortized variational inference for switching nonlinear dynamical systems." International Conference on Machine Learning. PMLR, 2020.

[4] Ansari, Abdul Fatir, et al. "Deep explicit duration switching models for time series." Advances in Neural Information Processing Systems 34 (2021): 29949-29961.

**Questions:**

- Are there any estimation guarantees of the proposed approach?
- Is it possible to determine the number of switching mechanisms, are the switches identifiable?

---

> ### Author Response · Authors · 2025-11-22
> **Thanks for pointing us to additional references.**
>
> We thank the reviewer for the positive remarks on the presentation of our work and for pointing us to these additional references. We have carefully reviewed the suggested papers and added a dedicated paragraph to the Related Works section (Section 1.1) explicitly citing and contextualizing this line of research. It was our oversight not to include these discussions in the initial submission.
>
> However, we believe some aspects of our model were misunderstood, and we respectfully disagree with the suggestion that our technical contributions are identical to or subsumed by prior work. To clarify the distinctions, we provide focused explanations and comparisons in terms of core modeling assumptions, dependency structure, the role of the Gumbel distribution, and inference compatibility. Notably, several cited works use Gumbel noise only as a differentiable sampling or reparameterization tool within the inference network and rely on specialized structured variational schemes, whereas GDM is a Gumbel-driven dynamical system whose latent evolution is directly governed by the Gumbel distribution and remains fully compatible with standard amortized BBVI. We provide point-by-point clarifications below in order of conceptual proximity to GDM, and we have incorporated corresponding revisions in both the main text and the appendix.

---

> > ### Author Response · Authors · 2025-11-22
> > **Thanks for pointing us to additional references (cont 1.)**
> >
> > **P1** *Becker-Ehmck et al. (2019) [2]: I believe this paper proposes exactly GDM (see Section 2.3 in their paper), for exactly the same type of SDSs (recurrent and linear). Please clearly articulate what is substantively new beyond those works. At present, I am not able to identify a principled difference between Becker-Ehmck et al. (2019) [2] and the proposed GDM; the submission presents GDM as novel, but it looks like a restatement of [2].*
> >
> > We appreciate the reviewer’s comparison with Becker-Ehmck et al. (2019). However, our model is structurally and conceptually distinct from theirs. Becker-Ehmck et al. (2019) use the Gumbel relaxation as a gradient-flow tool for an otherwise standard SLDS, whereas our work introduces a Gumbel-driven dynamical system, a structurally different, more interpretable, and computationally efficient model. The two approaches differ in both formulation and purpose:
> >
> > * Becker-Ehmck et al. remain within the SLDS framework, where the continuous latent dynamics are conditioned on a discrete switching variable. In their Section 4.1 (Eqs. 8–10), the authors simply replace the non-differentiable argmax over discrete states with a Gumbel–Softmax relaxation to enable gradient flow; the rest of the model is identical to a standard SLDS. As they explicitly note below Eq. 7, ``We do not condition the likelihood for the current observation directly on the switching variables. This limits the influence of the switching variables to choosing a proper transition dynamic for the continuous latent space”, meaning the discrete variable only selects a transition dynamic for the continuous latent space. In contrast, our formulation conditions the observations  directly on the switching variable,  removes the intermediate Gaussian latent layer, and introduces a direct dependency between the observations and the state dynamics. Consequently, our model is fully Gumbel-driven, with no intermediate Gaussian latent state whose stochasticity explains the data. We further show (lines 189-203) that removing the important observation-to-state dependency makes GDM equivalent to a three-level mixture SLDS with a deterministic intermediate layer, which is introduced purely for interpretability. Comparing the graphical representations (Figure 1, left in our paper and Figure 1b in Becker-Ehmck et al) provides an overview of some of these differences.
> > * We note that the Becker-Ehmck et al model is conceptually similar to the prototype we discussed in lines 204-209. We explicitly analyzed its weaknesses (line 206), mainly, the continuous and discrete dynamics compete to explain the data; and proposed a BBVI-based solution for completeness (Appendix D). We also implemented this variant early in development but found it unsatisfactory --- performing worse than GDM in both accuracy and speed, and scaling poorly to long, high-dimensional time series. Moreover, the inference procedure in Becker-Ehmck et al. requires structured splitting and alternating updates, whereas our approach supports joint sampling and avoids customized inference machinery. We note that Becker-Ehmck et al. did not release their code, making empirical comparison difficult and of limited value.
> > * The modeling objectives differ fundamentally. In Becker-Ehmck et al., the Gumbel variables are auxiliary reparameterization tools to improve differentiability and are not part of the generative process. Indeed, in Section 5 they report that the Gaussian formulation performs comparably or better. Importantly, Becker-Ehmck et al. does not report at all on the accuracy or interpretability of the inferred discrete variable. In contrast, we treat the Gumbel distribution as the driving noise of the dynamical system itself. GDM turns its heavy-tailed, extreme-value behavior into a way to modulate the stickiness and competition among states, leading to improved interpretability rather than only higher prediction accuracy. As we noted in our experiments (lines 441-443), a model can achieve high predictive accuracy while failing to recover meaningful states, indicating that trajectory fit alone is insufficient for interpretability.

---

> > > ### Author Response · Authors · 2025-11-22
> > > **Thanks for pointing us to additional references (cont 2.)**
> > >
> > > **P2** *Dong et al. (2020) [3]: This paper uses recurrent nonlinear SDS, it proposes exact inference on the switches ``collapsed", but in the experiments they also propose an additional method as baselines which uses Gumble-softmax relaxation in a similar way as GDM. See Appendix A.2 in their paper. Importantly, their amortized inference network uses a bidirectional RNN to encode the sequence and a forward RNN to produce per-time-step switch posteriors, which matches the inference architecture proposed in this paper for GDM. Please compare empirically to [3] on the proposed benchmarks and clearly articulate what is substantively new beyond Dong et al.’s amortized inference design.*
> > >
> > > We thank the reviewer for raising this comparison. We clarify below why Gumbel-Softmax SNLDS is conceptually and technically distinct from our work, and why an empirical comparison is neither feasible nor fruitful.
> > > * While amortized inference is standard, GDM is specifically designed to be compatible with off-the-shelf amortized BBVI. The strategy Dong et al. proposed relies on a specialized structured variational scheme involving a collapsed bound, alternating optimization of discrete and continuous latents, and annealing schedules. In contrast, GDM requires no customized derivations nor model-specific inference machinery and trains end-to-end with respect to its state dynamics. Ensuring that a new dynamical system remains compatible with generic amortized BBVI is a central design goal of our model, not a statement about novelty of amortization itself.
> > > * The role of the Gumbel distribution differs fundamentally. In Dong et al., the Gumbel–Softmax relaxation appears only in the variational posterior as a substitute for marginalizing over discrete states. It is not part of the generative model and does not define the switching dynamics. This makes their use of Gumbel comparable to Becker-Ehmck et al. (2019): the relaxation replaces the argmax inside the inference network only, not in the model. In contrast, GDM is a Gumbel-driven dynamical system in which Gumbel noise drives the switching process and determines the latent evolution. This is a structural, not methodological, difference.
> > > * The Gumbel-Softmax SNLDS variant in Dong et al. is structurally mismatched to its generative model. Their amortized posterior is mean-factorized so that the continuous latent posterior does not depend on the discrete state, even though in SNLDS the discrete state selects the transition dynamics. This breaks a core dependency in the model. This issue is reflected in their experimental results (Table 1): the Gumbel-Softmax SNLDS performs substantially worse than all other baselines including linear SLDS variants, despite SNLDS being strictly more expressive, empirically demonstrating that the proposed GDM is different in key aspects.
> > >
> > > In the absence of a public codebase, adapting Dong et al.'s structured inference procedure, collapsed objective, and annealing heuristics to new datasets would be challenging. Given the conceptual mismatch between Gumbel-Softmax SNLDS and GDM, and the experimental results reported in Dong et al., as explained above, we believe that a direct empirical comparison would not be fruitful. To address the reviewer's concern, we have cited the Dong et al. paper in the revised version of Section 1.1 and discussed the similarities and dissimilarities between GDM and all the literature mentioned by the reviewer in Appendix A.

---

> > > > ### Author Response · Authors · 2025-11-22
> > > > **Thanks for pointing us to additional references (cont 3.)**
> > > >
> > > > **P3** *Fraccaro et al. (2017) [1]: Kalman VAE. A Kalman filter in latent space, with a soft-switching mechanism. In their paper, they propose using a gumble-softmax sampling approach to estimate switches. Compare empirically to [1] on the proposed benchmarks and clearly articulate what is substantively new beyond those works.*
> > > >
> > > > Thanks for bringing this paper to our attention. We emphasize that GDM has a fundamentally different graphical structure from the KVAE of Fraccaro et al. (2017). As illustrated in Figure 1 of both papers, KVAE combines a VAE with a linear Gaussian state space model (LGSSM): observations are first mapped into a low-dimensional latent space by a deep neural network, and those latents are then explained by a soft mixture of LGSSMs. The dynamics-parameter network in KVAE (Section 3.3) corresponds conceptually to the dynamic-operator weighting in p-dLDS (Chen et al., 2024), where the model learns continuous weights over multiple linear dynamical components.
> > > >
> > > >
> > > > Crucially, KVAE does not impose any structural prior, sparsity constraint, or regularization on these mixture weights. As a result, the learned dynamics tend to be noisy mixtures rather than interpretable switching behaviors. This limitation is directly visible in our experiments: using the publicly released KVAE code, we evaluated the model on the NASCAR benchmark. The updated Table 1 shows consistently low discrete-state inference accuracy for KVAE compared to our model. Moreover, the updated Figure 1 illustrates an exempler dynamics-parameter network weighting derived from KVAE, in which the maximum state weights are below 0.30 throughout the time span.
> > > >
> > > >
> > > > Regarding the mention of Gumbel–Softmax in their Appendix D, the authors only state in one sentence that the dynamics-parameter weights could be "approximated as a discrete random variable using the Gumbel distribution.” This remark is not accompanied by any architectural modification, dependency analysis, or implementation. The paper does not specify how a discrete dynamics variable would interact with the KVAE structure or how inference would be performed. As in other prior works, the Gumbel distribution is referenced merely as a generic differentiable proxy for discrete sampling, not as a mechanism that defines the latent dynamics. **In contrast, GDM is a dynamical system explicitly driven by Gumbel noise, rather than a soft mixture or auxiliary inference trick.**
> > > >
> > > > **P4** *Ansari et al. (2021) [4]: Same as Dong et al. (2020), but the switches are explicit duration models.*
> > > >
> > > > We thank the reviewer for sharing this interesting work. However, its connection to GDM is conceptually weak. RED-SDS extends the SLNDS framework of Dong et al. (2020) by assuming nonlinear continuous dynamics and augmenting it with explicit-duration variables to model sojourn times for each discrete state. In contrast, GDM, in its dependency-removed case, deliberately retains linear dynamics at the continuous latent level to preserve interpretability and does not rely on explicit-duration assumptions.
> > > >
> > > >
> > > > The primary contribution of RED-SDS is the combination of state-dependent and time-dependent switching, where the latter is introduced through an additional latent variable that specifies the duration distribution for each mode. This imposes a strong parametric assumption on dwell times of the dynamical primitives. While this may be reasonable in some domains, it is inappropriate in others (e.g., the bouncing-ball benchmark in their paper), where switching is governed by state rather than state duration. More broadly, we pursue a model that infers switching structure from data rather than imposes explicit temporal constraints. In GDM, flexible switching behavior emerges naturally through the dependency between the current state and the previous observation, without committing to a specific sojourn-time model. A RED-style extension of GDM to incorporate explicit-duration switches could be an interesting direction for future work, but it lies outside the scope of our present contribution.
> > > >
> > > > As with the other papers mentioned by this reviewer, we have cited Ansari et al. in the revised version of our manuscript.

---

> > > > > ### Author Response · Authors · 2025-11-22
> > > > > **Thanks for the questions about the theoretical aspects. We have addressed the questions asked.**
> > > > >
> > > > > **P5** *The presented methodology lacks theoretical analysis on the presented methodology. Analysis on whether GDM is identifiable would be an important problem (i.e. the ground-truth representation is unique up to certain equivalence, e.g. permutaion of the switches). Without at least a statement of the assumptions under which GDM recovers meaningful modes, it is difficult to argue that the discovered “states” are interpretable rather than arbitrary mixtures. Are there any estimation guarantees of the proposed approach? Is it possible to determine the number of switching mechanisms, are the switches identifiable?*
> > > > >
> > > > > We appreciate the interest in formal guarantees regarding identifiability, estimation, and the determination of the number of switching mechanisms. We discuss each of the three aspects in the points below:
> > > > >
> > > > > * Following a recent work by Balsells-Rodas et al. (2024), we discuss the identifiability of GDM in the limit $\tau \to 0$. While the projection matrix $F$ and dynamic matrices $S_k$ are identifiable only up to an invertible linear transformation, the remaining parameters are identifiable up to permutations in the limiting case $\tau\to 0$. As we highlight in Appendix E, characterizing identifiability for the non-limiting case $\tau>0$ is substantially more difficult, because the relaxed latent process induces an infinite mixture over continuous state paths. A full extension of the finite-mixture theory to this setting is nontrivial and remains open.
> > > > > * Since GDM is trained using standard gradient-based optimization, the objective inherits the convergence guarantees of gradient descent under L-smoothness, a condition satisfied by our parametrization. Thus, every run is guaranteed to converge to a first-order stationary point (a local minimum or saddle point). We note that this provides stronger and more classical optimization guarantees than the popular variational Laplace EM (vLEM) algorithm (Zoltowski et al., 2020), whose  E-step does not generally guarantee an increase in the evidence lower bound (ELBO). While global optimality is not guaranteed (as is standard in nonlinear latent-variable models), the estimation guarantees for GDM follow the established theory for smooth nonconvex optimization.
> > > > > * The identifiability of the number of switching mechanisms depends on the considerations for over-specified mixture models. For example, if the true system has three dynamical primitives but GDM is fit with four, the model can represent two primitives as identical, in which case they are not identifiable. In this case, having more states than needed to explain data will result in overfitting. In practice, the number of switching mechanisms can therefore be selected using cross-validation or held-out likelihood. Additionally, post-hoc diagnostics can detect redundant states; for instance, by measuring similarity among the learned dynamic matrices $S_k$. If two $S_k$’s are nearly identical, the model effectively uses fewer than $K$ modes, allowing us to systematically reduce $K$. This behavior reflects a Bayesian Occam’s-razor effect: when more states are provided than needed, the unnecessary ones become unused or auxiliary compared to existing modes, which is also seen in our experiment results (Figure 3 for F1).
> > > > >
> > > > > While the present work is, to our knowledge, the first methodological study of dynamical systems with Gumbel-driven latent processes, we agree that a full theoretical characterization remains an important direction for future research. We thank the reviewer for highlighting this important avenue.

---

> ### Comment · Reviewer_gbmR · 2025-11-25
>
> I thank the reviewers for their time and efforts in their rebuttal. I first summarise my response and then provide more details.
>
> **Summary:** While I really appreciate the time dedicated to responding to my arguments and addressing the weaknesses that I consider critical for acceptance, I remain unconvinced on three fronts:
> - Novelty is incremental in comparison to previous works.
> - More empirical evidence is required for improvement claims.
> - Interpretability claims are not backed by identifiability
>
> **GDM in comparison to previous works**
>
> I provide what I understand is your model, and apologise if my previous claims with respect to Becker-Ehmck et al. (2019) came as too strong. I do believe there are some differences in estimation, but I consider the proposed models to be very close.
> - Generative model:
>   - Time series observations $y_t$, which are governed by Gumbel dynamics $z_t$ in latent space. Eq. (1)
>     - Transitions $y_t | y_{t-1}, z_t$: linear soft-mixing in terms of Gumbel variable $z_t$
>     - Gumbel dynamics with recurrence: $z_t | z_{t-1}, y_{t-1} $
>     - I initially got confused by the comparison to SLDS, but I understand that you only care about the model as presented in Eq. (1) (correct if wrong).
>   - When we compare to Becker-Ehmck et al. (2019), we find the following:
>     - Section 4.1 Generative model in their paper, Equation (10) is exactly your transition dynamics $y_t$ (soft mixing via Gumbel variable). The only difference is that Becker-Ehmck et al. (2019) operates this in latent space, which in my opinion is more efficient as it allows you to either augment the dynamics with higher-order moments, or perform dimensionality reduction, while keeping a shared fixed emission.
>     - Therefore I do not agree with the following: “our work introduces a Gumbel-driven dynamical system, a structurally different, more interpretable, and computationally efficient model”. From my perspective, it’s the same model but the dynamics in $x_t$ are absorbed as $Fy_t$ in your paper. There are a few differences, e.g., sticky prior, or inference, but structurally they appear the same model to me.
> - Inference:
>   - From my understanding, a bi-RNN computes embeddings $e_{1:T}$ from $y_{1:T}$, and then these embeddings are used to sample $z_t$ sequentially (correct if wrong).
>   - Here I agree that it differs from Bekcer-Ehmck et al. (2019).
>   - However, this inference idea is the same from Dong et al. (2020) and Ansari et al. (2021), but in this case they use it for their continuous latents.
>   - In comparison to Dong et al. (2020) the Gumble-Softmax SNLDS in Appendix A.2 proposes a very similar approximate posterior for the switches: a bi-RNN with sequential sampling. While it is true they also care about continuous latents, this does not change my arguments in terms of novelty in estimation.
>
> For these reasons, I believe the novelty front of this paper is not very strong. While I understand the rationale for GDM in terms of caring about pure Gumble-driven dynamics, I don’t see it as a novel methodology.
>
> In regards to related works, I would like to make some explicit comments on your rebuttal.
>
> > The Gumbel-Softmax SNLDS variant in Dong et al. is structurally mismatched to its generative model. Their amortized posterior is mean-factorized so that the continuous latent posterior does not depend on the discrete state, even though in SNLDS the discrete state selects the transition dynamics
>
> I believe this is acceptable as it is a variational approximation. This is my opinion, but I think that the reason for the Gumble-softmax baseline to underperform with respect to the collapsed (original) setup is the variance reduction you gain from marginalising the switches in training time. This is why I consider you should really be comparing your work in contrast to Dong et al. (2020) and Ansari et al. (2021).
>
> >  However, its connection to GDM is conceptually weak (comments on Ansari et al. (2021)
>
> >  More broadly, we pursue a model that infers switching structure from data rather than imposes explicit temporal constraints.
>
> I do not agree the connection is weak, as both models have the same objective, which is extracting switching modes to gain interpretable dynamics. Given that your paper aims to claim gains in interpretability, you should consider this recent model as a baseline. If GDM is a better fit than Ansari et al. (2021), then your model should outperform and you will be able to claim performance over other switching dynamical system baselines. This would substantially improve the significance of your model.

---

> ### Comment · Reviewer_gbmR · 2025-11-25
>
> **Comparisons to related work**
>
> I thank the reviewers for including KVAE in their experiments. However, I really think this work should compare against other SDS baselines, such as Dong et al. (2020) or Ansari et al. (2021). There is indeed a public codebases for these two papers:
>
> - Dong et al. (2020): https://github.com/google-research/google-research/tree/master/snlds
> - Ansari et al. (2021): https://github.com/abdulfatir/REDSDS
>
> If you want to experiment with a non-recurrent SDS, there is also this code from Balsells-Rodas et al. (2024) which you referenced in your rebuttal.
>
> https://github.com/charlio23/identifiable-SDS
>
> Even though in inference these models use a collapsed approach where they marginalise the switches, in test time it is possible to compute the switching posterior. Therefore, the models are able to perform segmentation and serve as baselines you should include in your work.
>
> **Comments on identifiability**
>
> This is probably not the most important point, but I am very happy to discuss identifiability of regime-switching models, such as GDM or SDSs.
> - From my perspective, identifiability is a key property that links likelihood to model solutions, i.e., 1-1 correspondence to number of switches, transitions. Therefore, to argue in favour of interpretability, comments on whether your system is identifiable are crucial. Without identifiability, we do not know how many switches our system has, ruining interpretability (as we could fit models with more or less switches that could provide contradictory interpretations without knowing which is the correct one.
> - In the limit $\tau \to 0$ you can borrow results from Balsells-Rodas et al. (2024) for GDM. However, be aware that your model allows recurrences from y_t, which are prohibited. There is an improvement where they allow arbitrary recurrence from the switches here Balsells-Rodas et al. (2025) [1].
> - For $\tau > 0$, it is true that we enter in the realm of infinite mixtures, and identifiability results do not extend directly. For this, I can recommend Brandorff-Nielsen (1965) [2], where they allow mixtures where the mixing integrand is continuous.
> - Critically, in the general case GDM cannot guarantee identifiability (for now). Therefore, I am not sure I can agree with the claims of GDM being more interpretable mentioned in the rebuttal or main text, as other SDS baselines are more likely to benefit from identifiability guarantees.
>
> > The identifiability of the number of switching mechanisms depends on the considerations for over-specified mixture models.
>
> In this case, identifiability results also aim to cover the number of switches, not only transition functions. This means that in estimation, we can treat the number of switches as a parameter to optimise (we usually do it by model selection), and we will pick the number of switches which maximises the likelihood (assuming estimator consistency).
>
> [1] Balsells-Rodas, Carles, et al. ‘Causal Discovery from Conditionally Stationary Time Series’. ICML 2025
>
> [2] Barndorff-Nielsen, Ole. "Identifiability of mixtures of exponential families." Journal of Mathematical Analysis and Applications 12.1 (1965): 115-121.

---

> > ### Author Response · Authors · 2025-11-26
> > **Thank you for your careful reading. We have further clarified the novelty of our approach.**
> >
> > **R1** *I provide what I understand is your model, and apologise if my previous claims with respect to Becker-Ehmck et al. (2019) came as too strong. I do believe there are some differences in estimation, but I consider the proposed models to be very close. I initially got confused by the comparison to SLDS, but I understand that you only care about the model as presented in Eq. (1) (correct if wrong). Section 4.1 Generative model in their paper, Equation (10) is exactly your transition dynamics  (soft mixing via Gumbel variable). The only difference is that Becker-Ehmck et al. (2019) operates this in latent space, which in my opinion is more efficient as it allows you to either augment the dynamics with higher-order moments, or perform dimensionality reduction, while keeping a shared fixed emission. From my perspective, it’s the same model but the dynamics in $x_t$ are absorbed as $Fy_t$ in your paper.*
> >
> > Thank you for the careful reading and for re-evaluating your earlier point - we really appreciate it. We believe the remaining confusion stems from treating an inference-side relaxation as equivalent to a generative-model redesign. Below we summarize the concrete distinctions and the consequences:
> >
> > * **Different dependency structure (generative graph)**. Becker-Ehmck et al. (2019) follows standard SLDS structure: discrete state $z_t$ selects a transition for a continuous latent, $x_t$ (with Gaussian noise) then generates observations. Gumbel is used as a relaxation/auxiliary tool to make inference differentiable. While for GDM, observations are conditioned directly on the discrete switching variable; there is no stochastic Gaussian latent trajectory explaining observations. The switching process itself (driven by Gumbel noise on the simplex) is the primary generative mechanism. These models are not equivalent up to a linear absorption via $F$. To "absorb $x_t$", a necessary condition would require $x_t$ to be fully determined by states before time step $t$. This is impossible to achieve by model of Becker-Ehmck et al. due to the intermediate Gaussian noise process for the continuous dynamics. Removing this latent noise process results in significant gains in model identifiability, inference efficiency and state interpretability, and we explain these further below.
> > * **Identifiability and ambiguity**. In SLDS, there is a trade-off: stochasticity in $x_t$ can explain variation that could alternatively be attributed to switching variable $z_t$. This creates further unidentifiability issues for their model. On the other hand, by removing the stochastic $x_t$
> > 		layer, GDM removes this competing source of noise so the Gumbel-driven switches explain the variability directly. This reduces the primary source of non-identifiability present in SLDS-style constructions.
> >
> > 		This lack of competition between generative noise sources makes GDM **at least** as identifiable as those approaches. A practical demonstration of this is the spurious switching observed in previous literature: previous models would switch back and forth between explaining data via the generative Gaussian noise and the generative Gumbel noise, producing rapid switching. All of our experiments demonstrate that GDM significantly alleviates this issue.
> > * **Inference and optimization differences**. We respectfully disagree that the Becker-Ehmck et al. model is more efficient than GDM. Becker-Ehmck et al. require structured inference (splitting, alternating updates) because gradients from reconstruction have difficulty flowing through the Gaussian transition and the dependency structure.  In contrast, there is no need to design $q(x)$ for our model, while all flexibilities including augmentation and dimensionality reduction in SLDS are retained, as we explain in the next point.
> > * **Expressiveness vs interpretability trade-off preserved**. SLDS variants that keep Gaussian $x_t$ are flexible in modeling trajectories, but that flexibility can mask the discrete structure we target. GDM intentionally places the modeling burden on Gumbel-driven switching (on the simplex), yielding interpretable state sequences while retaining flexibility through the projection $F$ and expressive transition structure. Allowing direct dependency between the observation and the switching states help the gradient flow directly through the state transition parameters, potentially helping improve state interpretability. This also leads to further structural differences between the 2-level GDM and the model proposed by Becker-Ehmck.
> > * **Empirical engineering evidence**. We implemented the SLDS-style 3-level mixture prototype (Appendix C). It performed worse than GDM in both state-recovery accuracy and computational speed, and it scaled poorly to long, high-dimensional time series. Becker-Ehmck et al. did not release code, which complicates apples-to-apples reproduction.

---

> > > ### Author Response · Authors · 2025-11-26
> > > **Thanks for the review. We have included additional empirical evidence to support our claims.**
> > >
> > > **R2** *However, this inference idea is the same from Dong et al. (2020) and Ansari et al. (2021), but in this case they use it for their continuous latents. In comparison to Dong et al. (2020) the Gumble-Softmax SNLDS in Appendix A.2 proposes a very similar approximate posterior for the switches: a bi-RNN with sequential sampling. While it is true they also care about continuous latents, this does not change my arguments in terms of novelty in estimation.*
> > >
> > > Thank you for the follow-up clarification. We agree that the form of amortized inference is used in prior works, including in Dong et al. (2020), Ansari et al. (2021), and other variational latent-sequence models. We do not claim novelty in this generic inference pattern. However, the novelty of GDM lies in the generative structure that makes such amortization fundamentally simpler.
> > >
> > > Unlike these prior models, GDM has no continuous latent trajectory that must be inferred. This eliminates the need to design or optimize a coupled variational posterior over $(x_{1:T}, z_{1:T})$. In contrast, prior methods require structured variational families, alternating updates, and straight-through estimators to handle the stochastic continuous dynamics and their dependence on the discrete switching variables. **In GDM, there is no need to approximate $q(x)$ and no latent-to-latent stochasticity that gradients must pass through. Because observations depend directly on $z$, gradients propagate cleanly through the transition logits, making the model fully compatible with unmodified off-the-shelf amortized inference.**
> > >
> > > As a result, the inference problem in GDM is materially different: the generative assumptions remove the main difficulty that motivates structured variational methods in SNLDS and related models. This is not merely reusing an existing inference scheme, but rather a change in the generative model that fundamentally alters what must be inferred.
> > >
> > > **R3** *This is my opinion, but I think that the reason for the Gumble-softmax baseline to underperform with respect to the collapsed (original) setup is the variance reduction you gain from marginalising the switches in training time. This is why I consider you should really be comparing your work in contrast to Dong et al. (2020) and Ansari et al. (2021). I really think this work should compare against other SDS baselines, such as Dong et al. (2020) or Ansari et al. (2021). There is indeed a public codebases for these two papers.*
> > >
> > > Thank you very much for sharing the public codebases and for raising this point. We agree that marginalizing over the discrete states can, in some settings, reduce gradient variance during training. However, this explanation alone cannot account for the substantial underperformance observed for GS-SNLDS relative to linear SLDS variants. On the other hand, our model outperforms linear SLDS variants to a large extent, empirically demonstrating that our model uses the Gumbel distribution in key different aspects.
> > >
> > > To further clarify this point, we evaluated the collapsed SNLDS implementation on our soft sticky NASCAR dataset. The results (0.45 ± 0.05) confirm that while the collapsed version avoids state collapse and improves over SLDS, it still inherits the fundamental limitations of SDS-based models: difficulty capturing smooth variable-speed transitions and accurately representing stochastic mixtures of overlapping states. Moreover, unlike our method, the switching variables in the collapsed SNLDS (and SLDS-style models) are not inferred directly through the variational inference procedure, but they are instead derived post-hoc from the inferred latent trajectories. These are precisely the regimes where our benchmark departs from classical switching assumptions. Thus, even with marginalization, SDS-style models remain fundamentally weak in the problem settings we evaluate. These results, along with expanded discussion, have been incorporated into Section 4.1 of the revised manuscript.
> > >
> > > Moreover, variance-reduction via marginalization does not directly speak to state interpretability, nor does it address the key issue we emphasize: in SDS-style approaches, continuous latent noise and discrete switches interact in ways that blur the state boundaries. GDM avoids this ambiguity by removing the continuous latent path entirely and allowing gradients to flow directly to the transition parameters.
> > >
> > > Regarding additional baselines, we appreciate the suggestion to include Ansari et al. (2021). Their RED-SNLDS model introduces valuable extensions, but these contributions target different goals (e.g., duration time modeling) and remain within the SDS generative structure. For this reason, and given the behavior observed in our new collapsed-SNLDS evaluation, it is expected that RED-SNLDS would face similar challenges on the soft-transition benchmarks we study.
> > >
> > > We hope that the new results help clarify the fundamental generative differences between GDM and SDS-style models.

---

> > > > ### Author Response · Authors · 2025-11-26
> > > > **Thanks for the comments on identifiability. We have provided a more detailed discussion of this aspect.**
> > > >
> > > > **R4** *From my perspective, identifiability is a key property that links likelihood to model solutions, i.e., 1-1 correspondence to number of switches, transitions. Therefore, to argue in favour of interpretability, comments on whether your system is identifiable are crucial. Critically, in the general case GDM cannot guarantee identifiability (for now). Therefore, I am not sure I can agree with the claims of GDM being more interpretable mentioned in the rebuttal or main text, as other SDS baselines are more likely to benefit from identifiability guarantees.*
> > > >
> > > > We fully agree that identifiability is fundamental to interpretability. Importantly, GDM is not less identifiable than existing SDS models; it inherits at least the guarantees available to SDS-type systems and removes a major source of non-identifiability present in SDS formulations. As noted earlier, SLDS and SNLDS models suffer from a well-documented confounding between stochasticity in the continuous latent path $x_t$ and stochasticity in the switching process $z_t$. GDM removes this intermediate latent-noises, thereby eliminating exactly this ambiguity. Structurally, this makes GDM closer to identifiable ARHMMs than to continuous-latent SDS variants.
> > > >
> > > > As discussed in our previous rebuttal, a complete identifiability theory for the non-limiting case is nontrivial. Nevertheless, the introduction of Gumbel noise does not create qualitatively new sources of non-identifiability relative to the Gaussian noise injected into continuous latents in SDS models. Recent work by Balsells-Rodas et al. (2024) support this viewpoint: non-identifiability in such systems arises from symmetry classes and model over-specification, not from the specific choice of noise distribution. We appreciate the additional references (Balsells-Rodas et al. (2025) and Barndorff-Nielsen (1965)) and have incorporated them to further elaborate on this aspect.
> > > >
> > > > Finally, we note that prior SDS work does not provide complete identifiability analyses in their original submissions; in fact, most SDS papers do not discuss identifiability at all. Our work is, to our knowledge, the first to explicitly address identifiability in Gumbel-driven switching models. Developing a full understanding of identifiability beyond the limiting regime remains an important and substantive direction for future research.

---

> > > > > ### Comment · Reviewer_gbmR · 2025-11-26
> > > > >
> > > > > Dear Authors,
> > > > >
> > > > > Thank you for your detailed response. I think we largely agree on the modeling side, where we diverge is on (i) how distinct is GDM from prior work and (ii) what can we claim about interpretability. I address your points by citing sentences from your comments below.
> > > > >
> > > > > > Becker-Ehmck et al. (2019) follows standard SLDS structure: discrete state $z_t$ selects a transition for a continuous latent, $s_t$ (with Gaussian noise) then generates observations. Gumbel is used as a relaxation/auxiliary tool to make inference differentiable.
> > > > >
> > > > > As mentioned, in Section 4.1, Eq. (10), the mixing comes from a Gumbel variable. Otherwise, this $s_t$ they introduce would be one-hot in these equations. I agree it comes from the reparametrisation, but in practise we could make the same arguments for GDM. Therefore, this core mechanism is shared with GDM.
> > > > >
> > > > > > Identifiability and ambiguity. In SLDS, there is a trade-off: stochasticity in $x_t$ can explain variation that could alternatively be attributed to switching variable $z_t$.
> > > > >
> > > > > This is primarily an estimation issue (generally for SNLDS) and not related to identifiability. In SLDS with Gaussian noise in transitions (this also applies for exponential family noise), the observed distribution is a mixture of Gaussians in the temporal domain since Gaussians are closed under affine transformations, and mixtures of Gaussians are identifiable if means are different [1]. Therefore, theoretically, there is no tradeoff in explaining switching behavior if you care about $p(y_t | y_{t-1}, z_t)$. These notions are explained in both Kivva et al (2022) [2] (non-temporal case) and Balsells-Rodas et al. (2024) [3] (temporal case). With equal covariances, you remain with an affine ambiguity on the continuous latents, which can be disentangled with regime-dependent covariances. There could be exceptions in the recurrent case, and Balsells-Rodas (2025) [4] results aim to address this issue, where the result applies to SLDS thanks to Gaussians being closed under affine transformations.
> > > > >
> > > > > This notion even extends to nonlinear SDSs, where you can identify switches if you have a piecewise linear decoder, and analytic latent dynamics (see Balsells-Rodas et al. (2024)). Therefore, referring to identifiability, it is not correct that $x_t$ and $z_t$ compete to explain variation. The problem you would have for SLDS is an affine ambiguity in continuous latents unless you assume regime-dependent noise.
> > > > >
> > > > > I believe the authors want to refer to this "competition between $x_t$ and $z_t$ as the "state collapse" issue in SNLDS outline both by Ansari et al. and Dong et al., which happens when you allow nonlinear transitions, such that one switch absobs all the regime-switching behaviors. This is an issue that comes in estimation purely, and has nothing to do with identifiability theory. I would like to stress that this issue does not happen in the linear case thanks to linearity.
> > > > >
> > > > > [1] Yakowitz, S. J., & Spragins, J. D. (1968). On the identifiability of finite mixtures. The Annals of Mathematical Statistics, 39(1), 209-214.
> > > > >
> > > > > [2] Kivva, Bohdan, et al. Identifiability of deep generative models without auxiliary information. NeurIPS 2022
> > > > >
> > > > > [3] Balsells-Rodas, C., et al. On the identifiability of switching dynamical systems. ICML 2024
> > > > >
> > > > > [4] Balsells-Rodas, Carles, et al. ‘Causal Discovery from Conditionally Stationary Time Series’. ICML 2025
> > > > >
> > > > > > This lack of competition between generative noise sources makes GDM at least as identifiable as those approaches.
> > > > >
> > > > > Identifiability is not guaranteed in GDM, as we discussed earlier that for $\tau > 0$ we enter the infinite mixture realm, and such statements need to be accompanied by formal proofs.
> > > > >
> > > > > >  Inference and optimization differences. We respectfully disagree that the Becker-Ehmck et al. model is more efficient than GDM.
> > > > >
> > > > > I have not suggested  Becker-Ehmck et al. inference is more efficient. What I mentioned is on the generative modelling side. I indeed believe SLDS are more efficient than GDM in treating time series. As I mentioned, the continuous latents in SLDSs allow you to achieve two things:
> > > > >
> > > > > 1.  Perform dimensionality reduction on the observations, and model dynamic transitions in latent space. This is very suitable for very high-dimensional time series such as videos.
> > > > > 2. Augment the latent dynamics with high-order moments, while keeping explicit first-order transitions. For this, GDM needs to explicitly incorporate higher-order dependencies in the design, whereas with SLDS we can overparametrise the latent space.
> > > > >
> > > > > The reason why I consider this to be more efficient than GDM is purely from the generative modelling side, not from an inference perspective. Furthermore, if $p(y_t | x_t)$ is a delta centred on $x_t$ (noiseless, identity emission), then Becker-Ehmck et al. generative model is GDM, which is the main novelty point the paper is pushing (as switches inference shares ideas from Dong et al. and Ansari et al.).

---

> ### Comment · Reviewer_gbmR · 2025-11-26
>
> > However, this explanation alone cannot account for the substantial underperformance observed for GS-SNLDS relative to linear SLDS variants. On the other hand, our model outperforms linear SLDS variants to a large extent, empirically demonstrating that our model uses the Gumbel distribution in key different aspects.
>
> I agree, because the problem in SNLDS is mainly "state collapse" and therefore increasing gradient variance in training makes this problem even worse. I can imagine Gumbel solutions being useful in the linear case as “state collapse” is not such an important issue anymore. This is why I consider it would make a very good point to compare collapsed approaches in the linear case.
>
> I appreciate the experiments with Dong et al. (2020). However, all the additional baseline comparisons have been reported on the Nascar dataset. I believe that claiming robust performance gains and superiority over baselines requires including them in other datasets (not just Nascar). Regarding Ansari et al. (2021), it is a baseline that allows you to extract switches, and it is one of the strongest ones. Therefore, it should be present in your experiments, especially if claims on performance gains thanks to Gumbel dynamics want to be made.
>
> As a note, I find it very surprising that Dong et al. (2020) is even outperformed by KVAE, since in their paper they report ~85% on Dubins path (a similar, but harder version on synthetic Nascar). Would it be possible to show a visualisation for the non-sticky version of the dataset?
>
> **Notes on identifiability**
>
> In general, I agree most of SDS papers do not handle identifiability, which is why I don’t consider the possible unidentifiability of GDM an issue on the paper. The problem is that I do not consider it possible to claim GDM is more interpretable over other SDS based approaches, when GDM loses identifiability under this reparametrised generative modelling perspective. Please see further comments below.
>
> The reason why GDM is not directly identifiable from Balsells-Rodas et al (2024) is that the mixture under Gumbel noise has the following form:
>
> $$p(y_t | y_{1:t-1}) = \int_{\Delta^{K-1}} p(z_{t} | y_{1:t-1}) p(y_{t} | y_{t-1}, z_{t})  dz_t $$
>
> where $p(y_{t} | y_{t-1}, z_{t})$ is Eq (1), second line in your paper. This infinite mixture form can resul in a non-identifiable model, and formal interpretability gains (by arguing recoverability of ground truth) should be accompanied by new mathematical statements. This is very different than a finite mixture model and unfortunately is not supported by Balsells-Rodas et al. (2024).
>
> - GDM only benefits from recent identifiability results when $\tau \to 0$, as stated in both your updated paper and rebuttal. As mentioned, for general $\tau$ we have infinite mixtures and identifiability here is less clear.
>
> - GDM is not closer to identifiable ARHMMs than to SDSs. Once again in SLDS with Gaussian noise, the observation distribution is Gaussian conditioned on $y_t, z_t$, therefore. Identifiability of the observed transitions $p(y_t| y_{t-1}, z_{t})$ and the number of switches is preserved.

---

> ### Author Response · Authors · 2025-11-30
> **Thank you for your detailed and thoughtful engagement with our work. We answer your questions on novelty and interpretability below.**
>
> We thank the reviewer for the thoughtful engagement and for helping us more clearly position our contributions relative to prior work. We are happy to hear that we were able to address the concerns on GDM's differences in terms of modeling. We are happy to continue the discussion on the remaining points:
>
> **Q1 How distinct is GDM from prior work?** *I agree it comes from the reparametrisation, but... this core mechanism is shared with GDM.*
>
> As the reviewer now appreciates, our underlying generative model is different from prior work. This leads to a different inference scheme compared to prior work. Together, these differences are substantial enough to produce experimental results that consistently and significantly favor GDM over prior approaches.
>
> To summarize:
> * Although GDM shares the use of the Gumbel distribution with Becker-Ecker et al., the resulting generative model is not equivalent:
>     + GDM directly links states to observations.
>     + It does not blur latent dynamics with Gaussian noise.
>     + It cannot be interpreted as a relaxation of any standard SDS model.
> * While our posterior construction contains a component reminiscent of Dong et al., our use of the Gumbel distribution is not restricted to variational inference, and the inference problem we solve is fundamentally different.
> * GDM is not simply A (Becker-Ecker et al.) plus B (Dong et al.); rather, it forms a new, simple yet practically powerful class of models that benefits from but is not reducible to these prior ideas.
>
> **Q2 What can we claim about interpretability?** *This is primarily an estimation issue...I believe the authors want to refer to ...allow nonlinear transition*
>
> We distinguish two notions of interpretability used in the literature. First, a practical, human-aligned notion asks whether inferred latent states are meaningful to practitioners (Doshi-Velez and Kim 2017), e.g., whether they align with expert labels or yield compact, actionable descriptions of behavior. This pragmatic view has been emphasized in prior work on interpretability evaluation and taxonomy. Second, interpretability is also related to the formal definition of identifiability which asks whether the latent representation can be recovered from observed data in a statistically well-posed way. Classic results on identifiability for finite mixtures and latent-class models provide the foundation for such formal claims, and recent work has begun to extend these analyses to switching dynamical systems (Balsells-Rodas et al 2023).
>
> In this paper, we report results for both perspectives: (i) empirical state-matching accuracy quantifies the practical alignment with expert labels, and (ii) Appendix E gives a first step towards formal identifiability statements for GDM in the limiting regime, building on recent identifiability theory for SDSs. Taken together, these two strands justify our claim that GDM improves interpretability in practice while acknowledging that a full identifiability theory for the non-limiting regime remains an important direction for future work.
>
> We are sorry for the confusion created in the previous response and acknowledge the reviewer’s perspective that these noises do not formally “compete to explain variance’’ in the sense of identifiability theory, but rather give rise to estimation-level problems. We use this term to refer to the general failure to recover meaningful states. Indeed, this includes the "state collapse" issue, and we note that this behavior can also appear in linear SDS models. More generally, this also includes spurious switching and cases where the continuous trajectory is accurately recovered while the inferred states accuracy is poor. While identifiability theory provides important constraints, as the reviewer correctly noted, the estimation problems remain a central, practically relevant obstacle, and GDM addresses these by design (direct state to observation link and Gumbel-driven transitions), improving interpretability in practice.
>
>
> **Q3 Identifiability issues** *Identifiability is not guaranteed...I agree most of SDS papers do not handle identifiability...The problem is that...*
>
> We appreciate the reviewer's point that the formal identifiability properties of GDM in the $\tau>0$ regime remain open, and we will revise the main text to ensure it is explicit that our interpretability claims refer only to the empirical, pragmatic sense (the first notion in Q2) rather than to the theoretical notion (the second notion in Q2).
>
> Our discussion on the Gumbel v.s. Gaussian noises were intended only to give intuition toward a possible extension of identifiability theory to the non-limiting regime. We agree with the reviewer that developing identifiability theory for the non-limiting regime will require new mathematical statements. This is an important direction for future work, and we share the reviewer’s view that such advances would help clarify the theoretical relationship between GDM and existing SDS formulations.

---

> > ### Author Response · Authors · 2025-11-30
> > **We group the remaining discussion points and address them below.**
> >
> > **Q4 Efficiency of SLDS** *What I mentioned is on the generative modelling side. I indeed believe SLDS are more efficient... As I mentioned, the continuous latents in SLDSs allow you to achieve two things: Perform dimensionality reduction... Augment the latent dynamics with high-order moments, while keeping explicit first-order transitions...*
> >
> > We would like to address a potential confusion regarding latent representations: GDM **does maintain** internal latent representations through the $F$ and $S_k$ matrices. Importantly, it does not inject continuous noise to this latent representation, unlike previous work. Thus, the dimensionality reduction advantage the reviewer raises also applies to GDM.
> >
> > GDM’s parameterization can already incorporate expressive sequence models or higher-order dependencies. Our work focuses specifically on interpretable switching structure, whereas continuous-latent approaches emphasize latent-trajectory expressiveness and dimensionality modeling — both are valuable and can be combined. We therefore view integrating Gumbel-driven switching with expressive continuous-latent parameterizations as a promising direction for future work, and we appreciate the reviewer’s suggestion along these lines.
> >
> >
> > **Q5 SNLDS experimental results** *I agree, because the problem in SNLDS... This is why I consider it would make a very good point to compare collapsed approaches in the linear case. However all the additional baseline comparisons have been reported on the Nascar dataset... Regarding Ansari et al. and it is one of the strongest ones... As a note, I find it very surprising that Dong et al. (2020) is even outperformed by KVAE. Would it be possible to show a visualisation for the non-sticky version of the dataset?*
> >
> > We agree with the reviewer that the linear setting is the most appropriate regime in which to study collapsed approaches. This is precisely why we emphasize the NASCAR benchmark: it is linear, widely used, and highlights the modeling distinction between soft, smooth transitions and hard discrete switches, where GDM departs from the SDS family. The F1 data involve a larger number of states, and because SDS-type models assume a single active dynamical primitive at each time step, we expect all SDS-based baselines to exhibit the same failure mode as rSLDS in this regime. The CalMS dataset is substantially larger and more challenging than both NASCAR and F1, and we use it primarily to demonstrate scalability and robustness of GDM, as well as a key tradeoff between expressiveness and interpretability.
> >
> > We have added results for collapsed SNLDS, which confirm that marginalization helps in some respects and reduces state collapse but does not address the fundamental modeling mismatch for soft, overlapping, and variable-speed transitions (Figure 2B): collapsed SNLDS still struggles to represent smooth, variable-speed transitions and overlapping-state mixtures. Regarding Ansari et al. (2021), we agree it is a relevant baseline. A careful reading of the model shows that, once the explicit-duration mechanism is removed, RED-SDS reduces to a recurrent SNLDS, i.e., it retains the same discrete, non-overlapping switching structure as rSLDS or collapsed SNLDS. For our benchmarks, we therefore do not expect additional modeling advantages over these baselines, since the expressiveness in RED-SDS comes primarily from the explicit-duration (ED) component. As discussed, the explicit-duration module specifies a duration distribution for each dynamical primitive and constrains when transitions can occur. This can be beneficial for some settings, but it does not help model smooth variable-speed transitions. At the same time, because it is rooted in the same SLDS modeling assumptions (discrete, non-overlapping modes, one active dynamical primitive per time step), we expect it to encounter similar difficulties in the soft-transition benchmarks. Nevertheless, we are open to incorporating a focused experiment in a revision if deemed useful.
> >
> > In the standard NASCAR setting, the underlying process exhibits recurrent-only switching dynamics; indeed, the recurrent-only rSLDS recovers the ground-truth trajectory almost perfectly. As a side note, collapsed SNLDS failed to reach high state accuracy in this case as it fails to capture the differences between the dynamics of the pink and grey segments. For the soft-sticky case, collapsed SNLDS does not achieve high state accuracy: although it benefits from stronger latent-dynamics parameterization, it inherits the discrete-only state representation of SLDS, which makes it less suited to capturing the soft and smoothly varying transition structure present in this dataset. In contrast, KVAE employs a continuous latent trajectory with smoothly varying dynamics weights, which aligns with the underlying generative assumptions of the soft-sticky NASCAR. This softness in the latent dynamics helps explain why KVAE outperforms collapsed SNLDS in this particular regime.

---

> > > ### Author Response · Authors · 2025-11-30
> > >
> > > We hope that the clarifications, manuscript revisions, and additional experimental results adequately address the reviewer’s concerns. We would be glad to provide any further explanation that may be useful. We sincerely appreciate the reviewer’s careful reading and constructive engagement.

---

### Official Review · Reviewer_wAs1 · 2025-11-04

**Soundness:** 3
**Presentation:** 3
**Contribution:** 3
**Rating:** 6
**Confidence:** 3

**Summary:**

This paper proposes the Gumbel Dynamical Model (GDM), a new interpretable framework for modeling complex, non-stationary time series. By introducing a Gumbel-Softmax relaxation of discrete latent states, GDM bridges the gap between interpretable switching dynamical systems and smooth, continuous state transitions. The model remains fully differentiable, allowing efficient optimization via standard gradient descent and enabling amortized variational inference that generalizes to unseen sequences without re-optimization. Empirical evaluations on synthetic (NASCAR), real-world (Formula 1 telemetry), and behavioral (CalMS21 mouse social interaction) datasets show that GDM achieves higher state interpretability and comparable or better predictive accuracy than strong baselines such as SLDS, rSLDS, and p-dLDS, particularly under stochastic and soft transition regimes.

**Strengths:**

The paper presents a technically sound and conceptually elegant contribution that enhances both interpretability and scalability of state-space models. The Gumbel-based relaxation is well-motivated and effectively mitigates spurious rapid switching common in discrete-state models. The proposed amortized inference framework is a practical strength, enabling efficient reuse across datasets and improving generalization without retraining. The experiments are extensive, diverse, and clearly demonstrate GDM’s advantages in recovering meaningful latent states and handling uncertainty. The writing is clear and thorough, with strong empirical grounding and transparent comparisons against relevant baselines.

**Weaknesses:**

While the paper is thorough and technically solid, its scope remains focused primarily on the SLDS family of models. It does not explore broader connections to other modern dynamical modeling frameworks—such as continuous-time approaches like Neural ODEs or more expressive deep sequence models—which could help position GDM within the wider landscape of time-series modeling. The discussion of the Gumbel relaxation, while mathematically correct, could be made more intuitive to clarify why interpretability is preserved under soft transitions. Additionally, the impact of the temperature parameter is discussed but not systematically explored or adapted in experiments.

**Questions:**

1) Have the authors considered learning or annealing the Gumbel temperature τ instead of fixing it, to better balance interpretability and flexibility?
2) Can the authors provide or suggest a quantitative metric to evaluate interpretability beyond qualitative visualizations?
3) How does GDM relate to or differ from continuous-time latent models (e.g., Neural ODEs), and could the approach extend to that setting?

---

> ### Author Response · Authors · 2025-11-22
> **Thanks for the constructive suggestions regarding the scope of our study.**
>
> We thank the reviewer for the positive assessment of the technical soundness, presentation, and novelty of our work, as well as for the constructive suggestions regarding the scope of our study. We have addressed these points in detail below and have revised the manuscript accordingly. Additional clarifications and updates have also been incorporated into the appendix.
>
> **P1** *Its scope remains focused primarily on the SLDS family of models. It does not explore broader connections to other modern dynamical modeling frameworks—such as continuous-time approaches like Neural ODEs or more expressive deep sequence models—which could help position GDM within the wider landscape of time-series modeling. How does GDM relate to or differ from continuous-time latent models (e.g., Neural ODEs), and could the approach extend to that setting?*
>
> We agree with the reviewer that positioning GDM within the broader class of dynamical models will provide better context and improve the exposition. Below we discuss some of the differences and how they can be combined. To address the reviewer's concern, we have revised the Related Work section accordingly, adding references to the broader literature, including continuous-time approaches.
>  * Our focus in this work is to describe a given complex time series in terms of interpretable switching dynamical primitives, for which SLDS-type models remain the most appropriate baselines. Although many modern architectures---including neural differential equations, RNNs, S4, and transformers---can achieve remarkable performance in sequence prediction and function approximation, they are generally not designed to produce switching latent dynamical primitives and temporal intervals. Since the central objective of our GDM is to discover meaningful switching structure while retaining predictive performance, the SLDS family provides the most relevant comparison class.
> * GDM is not restricted to the classical SLDS parameterization. As discussed in lines 179–184, the Gumbel-driven switching process can be parameterized by expressive sequence models such as RNNs, and the variational posterior can incorporate bidirectional RNNs. This leads to architectures reminiscent of e.g., LFADS (Pandarinath et al., 2018), but with a crucial difference: in GDM, these neural networks parameterize the transition logits of the Gumbel-driven switching process, whereas in LFADS they govern a continuous Gaussian latent trajectory. Thus, modern neural sequence components can be integrated into GDM without altering its relaxed-discrete, interpretable generative structure.
> * Our experiments underline a key trade-off: adding RNN/bidirectional RNN components improves observation-level accuracy but reduces interpretability of the inferred latent state level. We now include results for these variants in Figure 4 (for the CalMS dataset) to illustrate this expressiveness–interpretability trade-off and to clarify why our focus in this work is on interpretable state discovery.
> * Thank you for bringing up neural differential equations, and for asking how GDM can be related to this framework. Neural differential equations provide a principled way to model complex continuous-time dynamics in low-dimensional latent space, with successful applications in fields such as fluid dynamics and physics (Sholokhov et al., 2023) and systems neuroscience (Kim et al., 2021). While the dynamics function in these examples is parameterized by a multilayer perceptron (MLP), a continuous-time version of GDM would require a mechanism that governs the continuous-time evolution of the switching weights. Another key distinction is that GDM does not explicitly model the continuous-time evolution of the low-dimensional latent trajectory itself. Instead, GDM models how the relative weights over discrete primitives evolve over time, and these weights determine the resulting low-dimensional trajectory. In other words, GDM models dynamics over the probability simplex, sidestepping the need to model the dynamics of the time series itself, though these can be recovered directly from our model (please see line 200 and response to Reviewer gbmR for discussion on how this helps interpretability). This simplex structure also implies that the components of the switching variable must sum to 1 at all times. In short, the continuous-time analogue of this formulation would correspond to a neural stochastic differential equation with Gumbel–Softmax noise, making such an extension nontrivial. Nevertheless, developing such a continuous-time GDM could be a promising future direction, potentially bridging the gap between the expressiveness of neural differential equations and the interpretability of discrete switching models.

---

> > ### Author Response · Authors · 2025-11-22
> > **(Cont.) Thanks for your valuable suggestions!**
> >
> > **P2**  *The discussion of the Gumbel relaxation, while mathematically correct, could be made more intuitive to clarify why interpretability is preserved under soft transitions.*
> >
> > Thanks for your suggestion. The Gumbel is a member of the extreme-valued distributions family, and this property helps explain why interpretability is preserved even under a continuous relaxation. Extreme-value noise amplifies differences among competing logits, effectively sharpening the winner-take-all behavior behind the Gumbel–Max trick. Thus, although the relaxation is continuous, the resulting transition vectors remain close to one-hot, maintaining the semantics of discrete state switches while enabling smooth optimization. GDM then turns this heavy-tail, extreme-value behavior into a mechanism for modulating the stickiness and competition among latent states, preserving interpretable switching dynamics without enforcing a hard discrete constraint. We have added a version of this discussion under Section 2.1, which reviews the Gumbel-softmax trick.
> >
> > **P3** *Additionally, the impact of the temperature parameter is discussed but not systematically explored or adapted in experiments. Have the authors considered learning or annealing the Gumbel temperature $\tau$ instead of fixing it, to better balance interpretability and flexibility?*
> >
> > Thank you for this helpful suggestion. In the initial submission, we fixed the Gumbel temperature at $\tau=0.99$, which we found to be a practical choice: it yields transitions that are already very close to discrete while still ensuring stable optimization in PyTorch. In our experiments, lowering the temperature further  often led to optimization instability without noticeably improving interpretability. We agree that an annealing scheme can offer additional flexibility. We have now added an option to anneal the temperature during training, starting from a higher value and gradually reducing it to a target temperature near 1. Empirically, we observe that GDM is not particularly sensitive to the exact annealing schedule, and its performance is largely invariant within a broad temperature range, approximately $\tau\in (0.5,1)$, provided that training concludes at a reasonably low temperature (e.g., $\tau \approx 1$). This optional annealing mechanism is now documented and supported in our implementation.
> >
> > **P4** *Can the authors provide or suggest a quantitative metric to evaluate interpretability beyond qualitative visualizations?*
> >
> > Yes, we defined an inferred-state accuracy metric as a quantitative measure of interpretability, at the beginning of Section 4. This metric computes a state-matching score between inferred and true latent states, offering a simple yet meaningful quantification of interpretability in switching dynamical systems.

---

### Meta-Review · Area_Chair_Qjxt · 2026-01-08

**Summary:**

The paper introduces a Gumbel Dynamical Model, a continuous, Gumbel-based relaxation that enables smoother, non-stationary dynamics and fully differentiable, scalable training.

- paper positioning: mostly vs SLDS-family; lack broader context vs modern sequence models / continuous-time models

- temperature τ not systematically explored; asked about learning/annealing τ; strong dependence on τ suspected

- quantitative metric for interpretability beyond visuals; interpretability claim needs backup

- Scalability to high-dimensional observations.

- Empirical studies lack report of variability

- novelty - close to prior continuous relaxations for SDS and missing those citations.

**Reviewer Concerns:**

- novelty still disputed; reviewers argue GDM is “very close” to previous generative form and that the inference ideas resemble prior work

- Baseline coverage still insufficient; lack of multiple requested baselines

- Interpretability claims constrained; formal identifiability for τ>0 is open; interpretability claims as empirical rather than theoretical.

- new baseline experiments were mainly on NASCAR and need stronger evidence across other datasets and settings

**Reviewer Scores:**

Three reviewers were largely positive, with one negative reviewer, with substantial criticisms around novelty and baseline comparisons etc. The critical reviewers was not convinced after discussion and important concerns around novelty, empirical, interpretability claims remain all outstanding.

---

### Decision · Program_Chairs · 2026-01-26

Reject